# Targeting the Retinoblastoma/E2F repressive complex by CDK4/6 inhibitors amplifies oncolytic potency of an oncolytic adenovirus

Jana Koch [1,6,8], Sebastian J. Schober [2,8], Sruthi V. Hindupur [1,8], Caroline Schöning[2], Florian G. Klein [1], Klaus Mantwill[1], Maximilian Ehrenfeld [1], Ulrike Schillinger[1], Timmy Hohnecker[1], Pan Qi[1,7], Katja Steiger [3], Michaela Aichler[4], Jürgen E. Gschwend[1], Roman Nawroth [1] ✉ & Per Sonne Holm [1,5] ✉

CDK4/6 inhibitors (CDK4/6i) and oncolytic viruses are promising therapeutic agents for the treatment of various cancers. As single agents, CDK4/6 inhibitors that are approved for the treatment of breast cancer in combination with endocrine therapy cause G1 cell cycle arrest, whereas adenoviruses induce progression into S-phase in infected cells as an integral part of the their life cycle. Both CDK4/6 inhibitors and adenovirus replication target the Retinoblastoma protein albeit for different purposes. Here we show that in combination CDK4/6 inhibitors potentiate the anti-tumor effect of the oncolytic adenovirus XVir-N-31 in bladder cancer and murine Ewing sarcoma xenograft models. This increase in oncolytic potency correlates with an increase in virus-producing cancer cells, enhanced viral genome replication, particle formation and consequently cancer cell killing. The molecular mechanism that regulates this response is fundamentally based on the reduction of Retinoblastoma protein expression levels by CDK4/6 inhibitors.

The entry of oncolytic viruses into clinical application opens up groundbreaking changes in current and future treatment regimens. Oncolytic viruses (OV) based on adenovirus type 5 (Ad) belong to the best-characterized oncolytic virus drugs with high safety profiles in clinical application[1,2]. Despite their potent anti-cancer activity, clinical studies revealed limitations of currently used viral vectors and treatment regimens as monotherapy[3]. One of them is reduced replication of OV in target cancer cells, resulting from genetic modifications within the viral genome to gain tumor specificity. However, due to the low toxicity profile, the development of combination therapies that support replication of OV might just be a key to overcoming those challenges[4,5].

Over the past decades, studies on cell cycle regulation have revealed a central role of the CDK4/6-RB/E2F pathway in controlling

[1]Department of Urology, Klinikum rechts der Isar, Technical University of Munich, Munich, Germany. [2]Department of Pediatrics, Children's Cancer Research Center, Kinderklinik München Schwabing, School of Medicine, Technical University of Munich, 80804 Munich, Germany. [3]Department of Pathology, Klinikum rechts der Isar, Technical University of Munich, Munich, Germany. [4]Helmholtz Zentrum München, German Research Center for Environmental Health, Research Unit Analytical Pathology, Munich, Germany. [5]Department of Oral and Maxillofacial Surgery, Medical University Innsbruck, A-6020 Innsbruck, Austria. [6]Present address: Dr. Margarete Fischer-Bosch Institute for Clinical Pharmacology, Stuttgart, University of Tübingen, Tübingen, Germany. [7]Present address: Department of Urology, Shanghai General Hospital, School of Medicine, Shanghai Jiao Tong University, Shanghai, China. [8]These authors contributed equally: Jana Koch, Sebastian J. Schober, Sruthi V. Hindupur. ✉e-mail: roman.nawroth@tum.de; per-sonne.holm@i-med.ac.at

the transition from G1 to S-phase[6]. At the molecular level, it is widely accepted that Retinoblastoma protein (RB) restricts progression from G1 to S-phase by binding to and suppressing E2F transcription factors. Active RB is phosphorylated by cyclin-dependent kinase 4/6 (CDK4/6) and cyclin E/CDK2, leading to E2F release (free E2F) and subsequent G1 exit[7,8]. In the hypophosphorylated form, RB thus acts as a transcriptional repressor by binding to E2F[9], which can be reversed by adenoviral E1A[10]. Since this pathway is often deregulated in cancer, several inhibitors directed against CDK4/6 (CDK4/6i) have been developed to prevent phosphorylation of RB[11]. The dominant effect of CDK4/6i as monotherapy is the induction of cell cycle arrest in G0/G1 or senescence but not apoptosis[12–14]. In addition, treatment with CDK4/6i is often accompanied by a reduction of RB and E2F1 expression level[15–17]. In this regard, it was shown that E2F overexpression bypasses cell cycle arrest mediated by CDK4/6 inhibition[18]. Currently, three CDK4/6 inhibitors (Palbociclib (PD), Ribociclib (LEE), and Abemaciclib (LY)) have been approved by the FDA for the treatment of breast cancer. A limitation of these inhibitors is their low efficacy as a monotherapy that allows their clinical use only in combination with other drugs such as hormone ablation therapies[19]. However, due to their low toxicity profile, CDK4/6i are excellent drugs for combination with other therapies[11,13].

Adenoviral replication has been examined for decades[20] and is tightly linked to an interplay with the viral master regulator protein E1A, one of the first viral genes expressed after infection[21], and the cellular proteins retinoblastoma protein (RB) and E2F[22]. One key event in the adenoviral life cycle is the targeting of the RB-E2F interactions by E1A and activation of E2F transcription factors[23–25]. Consequently, E1A induces the simultaneous transition of infected cells into the S-phase, which is supposed to be an essential step in creating an appropriate environment for the replication of viral DNA and the activation of all other viral genes[26]. Both events are tightly regulated by E2F1, which regulates adenoviral E2-expression by binding to the two E2F-binding sites arranged in a palindrome in the E2-early promoter[27]. In addition, E2F also binds to the E1A-enhancer region and in turn activates the transcription of E1A[28]. Deletion of the E1A amino acids responsible for RB binding has been used to create promising oncolytic viruses that use free E2F to drive viral replication in cancer cells[29–31]. However, the importance of the RB/E2F-repressor complex on the viral life cycle has not yet been studied in detail. Besides E2F-1, which binds to the E2-early promoter, the human transcription factor YB-1 also facilitates E2-expression by binding to the E2-late promoter[32]. Upon adenovirus infection, YB-1 is translocated into the nucleus via the early viral proteins E1B55k/E4orf6. Deletion of the CR3-transactivation domain of the large E1A protein created the oncolytic virus, named XVir-N-31, which replicates in cancer cells displaying nuclear YB-1 expression without affecting the natural binding capacity of E1A to the RB protein[33].

In this study, we evaluate the antitumor effect of XVir-N-31 and adenovirus wild type (ADWT) in combination with different CDK4/6i in vitro in a panel of bladder cancer and sarcoma cell lines and in murine xenograft sarcoma mouse models. Our data show that in spite of the opposing roles in cell cycle regulation, a combination of the oncolytic adenoviruses and CDK4/6i acts synergistically by increasing viral genome replication and cancer cell lysis. Mechanistically, degradation of RB by CDK4/6i leads to a reduction of the transcriptional repressor complex RB/E2F, which results in an earlier and more efficient expression of virus-related genes such as E1A. In addition, transcriptional activation and stabilization of E2F1 by E1A contribute to the observed synergistic effect. These observations are confirmed in vivo, indicating an increased therapeutic effect of XVir-N-31 on tumor growth also in 3-dimensional murine model systems. The improved lysis of tumor cells might consequently result in better recognition of the tumor by the immune system and thus induces a systemic immune response against the tumor which is essential for the treatment of metastatic disease.

## Results

### Checkpoint kinase 1 inhibitors do not affect adenoviral genome replication

In a recent report, inhibition of ataxia telangiectasia and Rad3-related checkpoint kinase 1 (ATR-Chk1) was shown to enhance oncolytic toxicity of adenovirus in ovarian cancer by using the small molecule inhibitor UCN-01[34]. We explored this approach in bladder cancer cells and first examined infectivity and the relationship between MOIs used and cell lysis (Supplementary Fig. 1a, b)[35]. For the examination of combination therapies, we used an MOI that resulted in cell lysis rates of 10–30% throughout this manuscript. We treated cells using UCN-01 and the more specific Chk1 inhibitor AZD7762 observing that UCN-01 but not AZD7762 synergistically increased the toxicity of XVir-N-31 in T24, RT112, and UMUC3 cells (Fig. 1a, b). This effect was not observed in the RB-negative cell lines 639V and 647V, indicating that the presence of the RB protein is involved in the observed effect (Fig. 1c). These results suggested that inhibition of ATR-Chk1 may not play a key role in the observed combinatorial effect. Thus, we performed immunoblots on whole cell lysates in order to identify proteins that might be modulated only by UCN-01 but not by AZD7762 treatment. As shown in Fig. 1d, AZD7762 modified only Chk1 expression level which is consistent with other studies[36] while UCN-01 additionally down-regulated total RB and pRB levels at concentrations >20 nM without affecting E2F expression (Fig. 1d). UCN-01, in contrast to AZD7762, induces G1-S phase arrest in the cell cycle[36,37]. This off-target phenomenon of UCN-01 has been documented before as it also targets several other CDKs, AKT, and protein kinase C[38,39].

Since the RB/E2F complex plays a key role in the viral life cycle and the progress of cells into S-phase has been accepted as a key event in adenovirus replication, these results prompted us to investigate viral genome replication under serum starvation, which has been described to cause RB inhibition and G1 arrest[40]. This treatment strategy induced a strong increase in viral genome replication in infected T24 cells (Fig. 1e). Western blot analysis revealed down-regulation of pRB and total RB prior to infection (Fig. 1f), suggesting the involvement of the RB/E2F-axis in the observed effect.

### CDK4/6 inhibitors synergize with adenovirus genome replication

Suppression of RB expression level and induction of G0/G1 arrest is also induced by CDK4/6i, which also display very high specificity for their molecular target and are approved for clinical application[16]. Thus, we tested the three CDK4/6 inhibitors PD, LEE, and LY in combination with ADWT and XVir-N-31 on different bladder cancer cells and demonstrated a remarkable synergistic effect of combined treatment with XVir-N-31 or ADWT on cell lysis in T24, RT112, and 253J cell lines (Fig. 2a, Supplementary Fig. 1c). CDK4/6i do not induce apoptosis in these cell lines (Supplementary Fig. 1d, e). We next examined in a dose-dependent kinetic the correlation of PD treatment on RB expression level and cell lysis and could show, that with the molecular response to PD an increase in cell killing was induced by XVir-N-31 (Fig. 2b, c). Thus, the improvement of virus-induced cell lysis correlates with the expression level of RB but it does not require complete elimination of this protein. We then addressed the question of whether the time point of pretreatment with CDK4/6i had an impact on the enhanced oncolytic activity and did not observe any obvious differences between 24 h pretreatment with CDK4/6i before infection or parallel treatment (Supplementary Fig. 1f). This indicates that the improved therapeutic effect is not solely based on the initial synchronization of cells in G0/G1. The effects observed above were then correlated with viral genome replication and viral particle formation. Even at early time points, replication of the viral genome was significantly enhanced in combination with all three tested CDK4/6i, indicating that the observed oncolytic effects are based on increased genome replication (Fig. 2d,

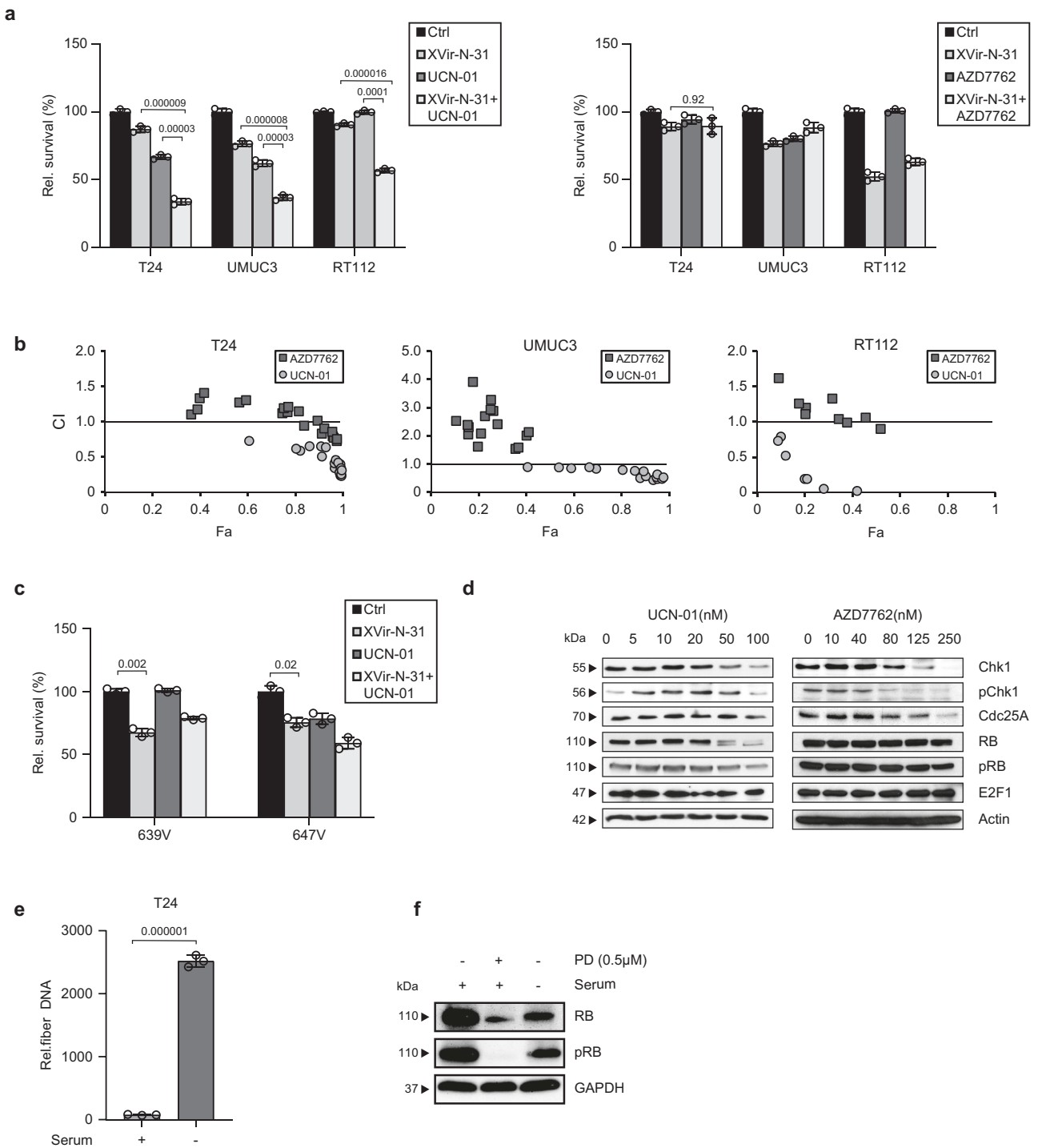

**Fig. 1 | Combination therapies that downregulate RB expression enhance oncolytic virotherapy. a** Cell proliferation analyses in bladder cancer cell lines. Bladder cancer cells were pre-treated with UCN-01 for 24 h and infected with XVir-N-31 (RT112 (40 nM, MOI 400), T24 (20 nM, MOI 40), and UMUC3 (20 nM, MOI 20)). T24, UMUC3, and RT112 were pre-treated with AZD7762 (80 nM) and infected with MOI 40, MOI 20, and MOI 400 of XVir-N-31, respectively. Cell viability was assessed at 4 dpi. Data are shown as the percentage of surviving cells (*n* = 3, mean ± SE relative to non-infected, untreated control. **b** Combination index (CI) plots show synergistic effects on cell viability in combination with UCN-01 (circles) but not with AZD7762 (squares). CI values were calculated using the Chou−Talalay theorem. CI < 1 synergism; CI = 1 additivity; CI > 1 antagonism; Fa: fraction affected. **c** RB-negative 647V and 639V cells were pre-treated with UCN-01 (20 nM) for 24 h and infected with MOI 2 and MOI 200 of XVir-N-31 respectively. Cell viability was

assessed at 4 dpi. Data are shown as the percentage of surviving cells (*n* = 3, mean ± SE) relative to non-infected, untreated control. *p* < 0.05 **d** Protein expression was analyzed by western blotting in T24 cells treated with increasing concentrations of UCN-01 or AZD7762 for 24 h. One representative blot is shown in three independent experiments. **e** Viral genome replication was assessed in serum-starved T24 cells infected with XVir-N-31 (MOI 50) by qPCR to amplify viral fiber DNA at 24 hpi. Data are represented relative to fiber DNA at 4 hpi as the baseline (*n* = 3, mean ± SD). **f** Protein expression was analyzed by western blotting in T24 cells that were serum starved for 24 h. One representative blot is shown in two independent experiments. SD standard deviation, SE standard error, hpi/dpi hours/days post-infection, MOI multiplicity of infection. The statistical significance was determined by a two-sided Student's *t*-test. *n*: number of biologically independent samples. Source data are provided as a Source Data file.

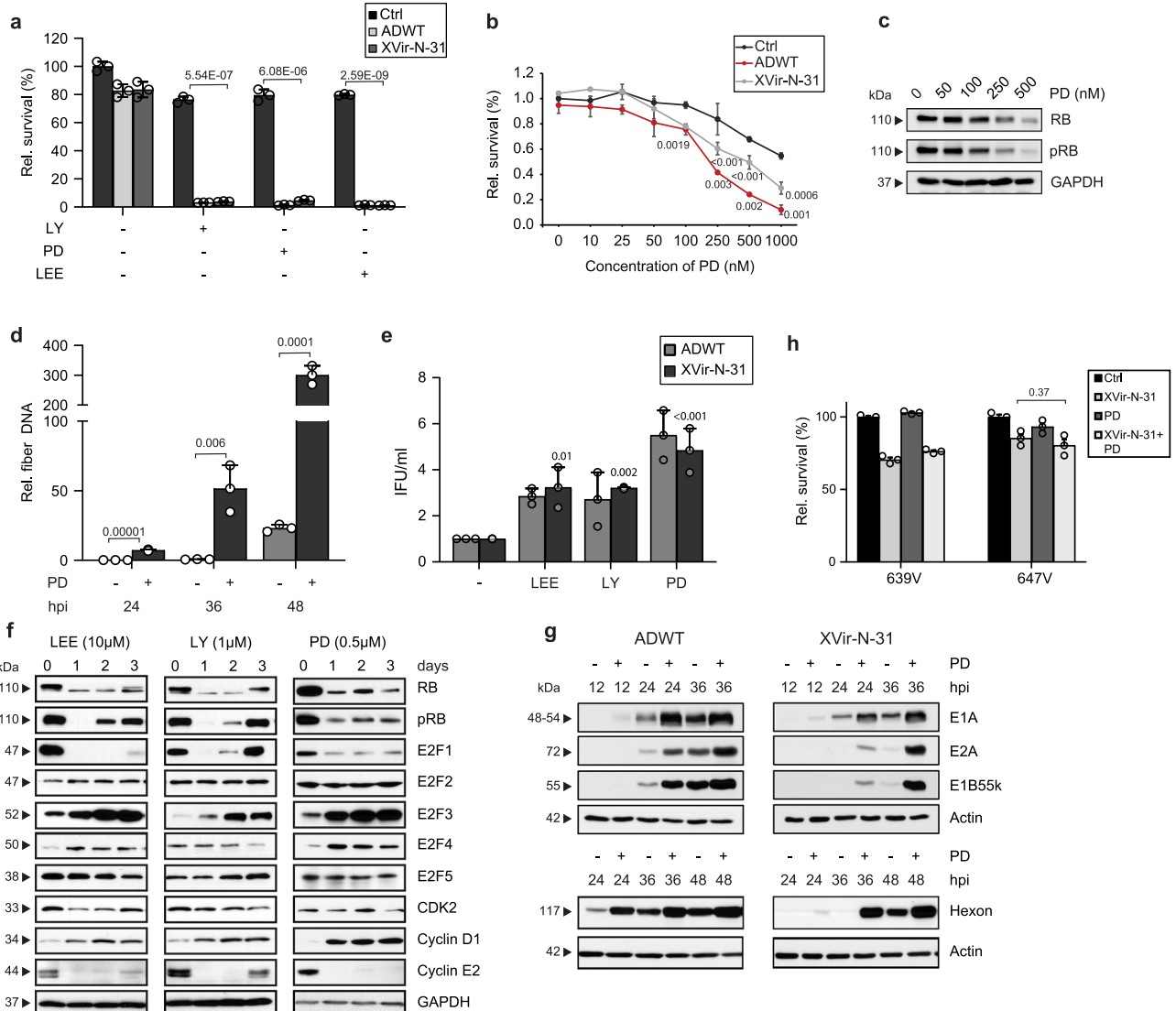

**Fig. 2 | Combination with CDK4/6 inhibition enhances oncolytic virotherapy.**
**a**, **b** Cell proliferation analyses in RB positive T24 cells. **a** T24 cells were pre-treated for 24 h with CDK4/6 inhibitors as indicated and infected with ADWT (MOI 80) or XVir-N-31 (MOI 60). **b** T24 cells were treated with increasing concentrations of Palbociclib for 24 h and infected with ADWT (MOI 50) and XVir-N-31 (MOI 50). Cell viability was assessed at 4 dpi. Data are shown as the percentage of surviving cells ($n = 3$, mean ± SE) relative to non-infected, untreated control. $p < 0.05$. **c** RB protein expression was analyzed in T24 cells upon treatment with increasing concentrations of Palbociclib for 24 h by western blotting. **d** Viral genome replication was assessed by qPCR to amplify viral fiber DNA in T24 cells pre-treated for 24 h with Palbociclib and infected with XVir-N-31 (MOI 50). Data are represented as relative fiber DNA at indicated time points compared to fiber DNA at 4 hpi as baseline ($n = 3$, mean ± SD). **e** Quantification of viral particles was performed by a hexon titer test in T24 cells that were pre-treated with indicated CDK4/6 inhibitors (5 μM LEE, 0.5 μM LY, and 0.5 μM PD) and infected with MOI 50 of ADWT or XVir-N-31. The titers are

presented as infectious units per milliliter (IFU/ml). $n = 3$, mean ± SE. **f** Cellular cell cycle protein expression was determined by western blotting in T24 cells treated with PD, LY, or LEE for up to 3 days. **g** Viral protein expression was analyzed at indicated time points by western blotting in T24 cells pre-treated with Palbociclib and infected with the viruses ADWT and XVir-N-31 (MOI 50). One representative blot is shown in three independent experiments. **h** Cell proliferation analyses were performed in RB-negative 647V and 639V cell lines pretreated with Palbociclib (2 μM) and infected with MOI 2 and MOI 200 of XVir-N-31, respectively. Cell viability was analyzed at 4 dpi. Data are shown as a percentage of surviving cells ($n = 3$, mean ± SE) relative to non-infected, untreated control. hbi hours before infection, hpi/dpi hours/days post-infection, SD standard deviation, SE standard error, MOI multiplicity of infection. The statistical significance was determined by a two-sided Student's $t$-test; $n$ number of biologically independent samples. Source data are provided as a Source Data file.

Supplementary Fig. 1g, h) and particle formation (Fig. 2e, Supplementary Fig. 1i).

CDK4/6i interfere with the expression level of a number of cellular proteins involved in the adenovirus replication, including E2F family members and RB. The expression of E2Fs and RB are essential elements in adenovirus replication and are involved in regulating cell cycle progression. Therefore, we studied effects in a time kinetic using the three CDK4/6i PD, LY, and LEE on protein expression levels in immunoblots (Fig. 2f). RB and E2F1 protein levels were largely downregulated after treatment with a slight recovery from day 2

onwards. However, E2F2-E2F5 was not downregulated or even upregulated upon CDK4/6i. As expected, Cyclin D1 level was upregulated and Cyclin E was downregulated as previously described[16]. Also, an increase and earlier onset of expression of viral transcripts and proteins (E1A, E2A, and hexon) were observed upon CDK4/6i treatment (Fig. 2g, Supplementary Fig. 1j). It is established that cancer cells with malfunctioning RB pathways are mostly resistant to the treatment with CDK4/6i[17]. When testing the combination therapy in RB-negative bladder cancer cell lines 639V and 647V, no positive effects on cell survival were observed, indicating again that this

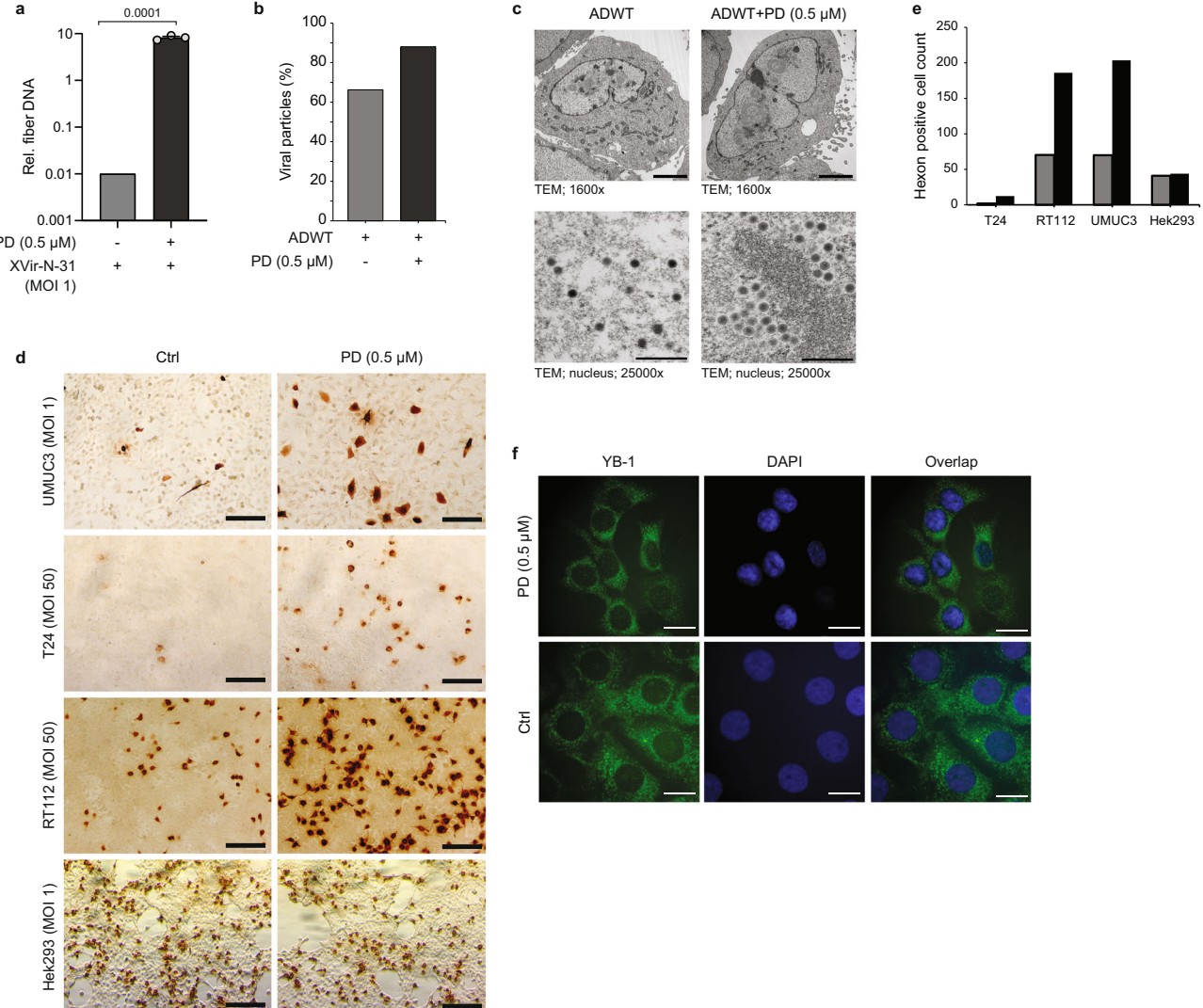

**Fig. 3 | CDK4/6i induces a cellular environment that augments virus replication. a** Viral genome replication was assessed by qPCR to amplify viral fiber DNA in T24 cells pre-treated for 24 h with Palbociclib and infected with XVir-N-31 (MOI 1). Data are represented as relative fiber DNA at indicated time points compared to fiber DNA at 4 hpi as the baseline (*n* = 3, mean ± SD). **b** Quantification of viral particles in electron microscopy in two independent samples. Data represent the mean of two experiments. **c** Representative electron microscopy images showing viral proliferation at 48 hpi in T24 cells pre-treated with Palbociclib and infected with ADWT (MOI 50). Scale bars in upper and lower panels represent 4 μm and 400 nm, respectively. **d** Representative immunocytochemistry images depicting hexon staining at 48 hpi in UMUC-3, T24, RT112, and Hek293 cells infected with MOI 1, 50, 10, and 1 of ADWT respectively. Scale bars represent 100 μM. **e** Quantification of hexon-positive cells from immunofluorescence. **f** Representative immunofluorescence images depicting staining against YB-1 protein in T24 cells treated with Palbociclib for 24 h. Scale bars represent 10 μM. SD standard deviation, hpi hours post-infection, MOI multiplicity of infection. The statistical significance was determined by a two-sided Student's *t*-test; *n* number of biologically independent samples. Source data are provided as a Source Data file.

combination treatment requires an active RB-E2F-signaling pathway in the cells (Fig. 2h).

## CDK4/6i induces a cellular environment that enhances virus replication

Key aspects in the clinical application of oncolytic viruses are not only viral spread within the tumor[41] and immune activation[42], but also the infectivity of cancer cells. This requires a certain threshold to initiate viral replication, which is a challenge to achieve in vivo and often hampers response to therapy. Therefore, we asked whether treatment with PD also affects viral genome replication at low MOI. As shown in Fig. 3a, treatment with PD enabled viral genome replication of XVir-N-31 at MOI 1 which otherwise shows no replication in non-treated T24 bladder cells (values below 1 indicate no viral replication). Thus, treatment of cells with CDK4/6i creates a highly favorable cellular environment for viral genome replication and has even the ability to

convert XVir-N-31 at very low MOI from a non-replicative into a replication-competent status in cancer cells.

In order to examine this effect in more detail, we performed an immunohistochemical and electron-microscopy analysis of infected cells. Both methods confirmed a significant increase of virus-producing cells at low MOIs after CDK4/6i treatment (Fig. 3b, c). This result was extended to three different bladder cancer-derived cell lines (UM-UC-3, T24, RT112) and Hek293 cells as a control since they do not respond to CDK4/6i although they express RB protein (Supplementary Fig. 2a). In the three bladder cancer cell lines, treatment with PD resulted in a dramatically increased number of virus-producing cells (Fig. 3d). This could also be confirmed by quantification of hexon positive cells (Fig. 3e).

To rule out that increased infectivity was the reason for the observed effect, we assessed the abundance of viral genomes at 4 h post-infection. It has been shown that cellular adenovirus receptor

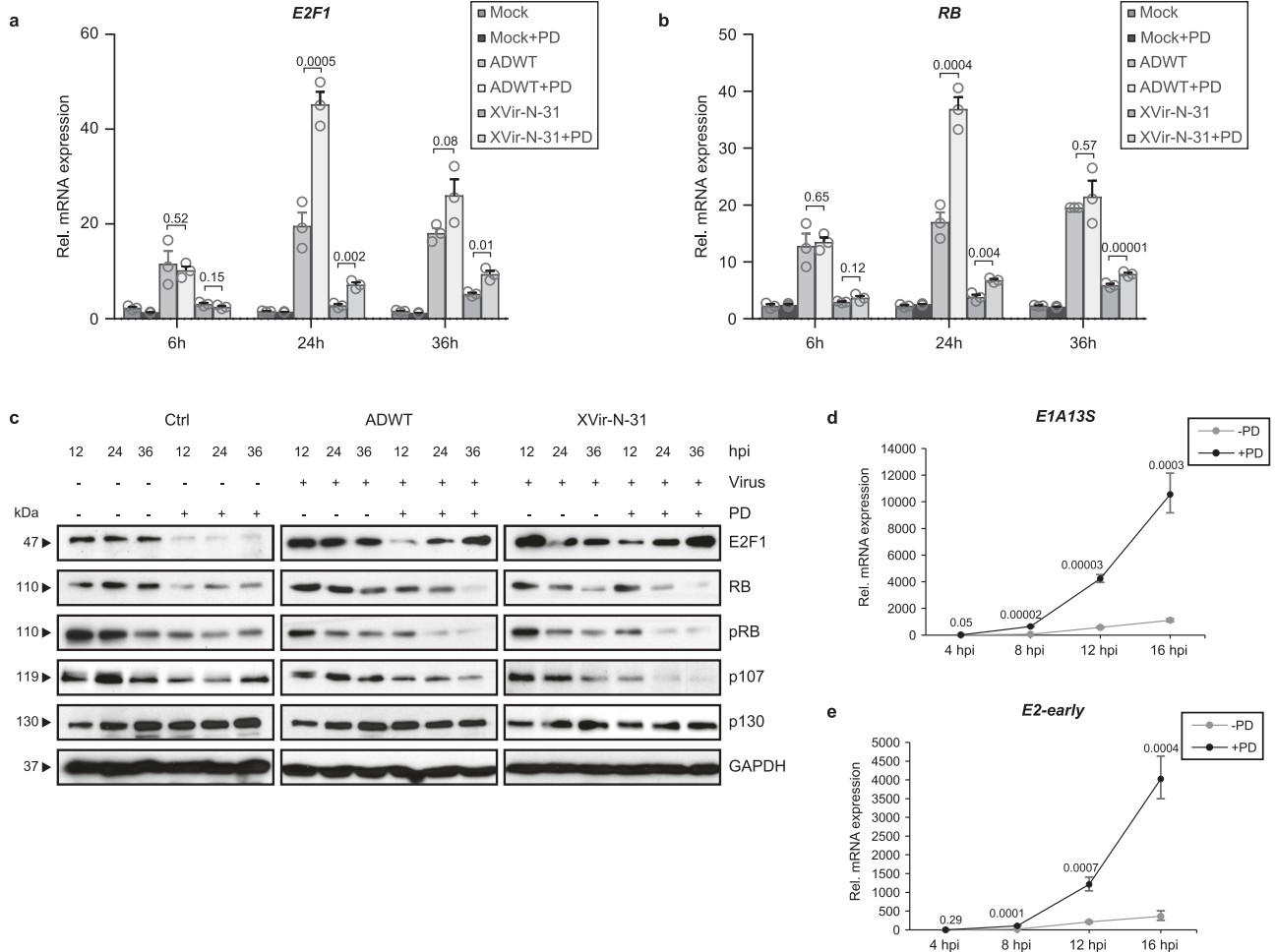

**Fig. 4 | The modulation of RB–E2F protein complex by CDK4/6i is responsible for enhanced viral genome replication.** T24 cells were pre-treated for 24 h with 0.5 μM Palbociclib and infected with ADWT or XVir-N-31 (MOI 50). **a**, **b** Cellular gene expression was analyzed by qRT-PCR at indicated time points for the genes E2F1 (**a**) and RB (**b**). Data are presented as relative mRNA expression compared to the housekeeping gene beta-actin (*n* = 3, mean ± SD). **c** Protein expression was analyzed at indicated time points by western blotting. One representative blot is shown in three independent experiments. **d**, **e** Viral gene expression was analyzed in ADWT-infected cells by qRT-PCR at indicated time points for the E1A13S gene (**d**) and E2-early transcripts (**e**). The data shown represent the mRNA expression in treated samples relative to Actin at each time point (*n* = 3, mean ± SD). SD standard deviation, MOI multiplicity of infection. The statistical significance was determined by a two-sided Student's *t*-test; *n* number of biologically independent samples. Source data are provided as a Source Data file.

(CAR) is not changed in its expression level upon PD treatment CAR[43,44] a finding we could confirm 8 and 24 h after treatment with PD (Supplementary Fig. 2b). As for viral replication, we could not observe a difference of infection after treatment with CDK4/6i 4 h post-infection (Supplementary Fig. 2c), indicating that infectivity is not the critical factor for increased potency of Ad vectors after CDK4/6i treatment.

Next, we analyzed the influence of CDK4/6i on YB-1 as a cell cycle-regulated factor, that plays an important role in the adenovirus life cycle by inducing the expression of AdV DNA polymerase (AdV Pol). Immunofluorescence staining and western blot analysis of YB-1 showed no increased expression or translocation of YB-1 into the nucleus upon CDK4/6i treatment, indicating that YB-1 is not directly involved in this combination therapy (Fig. 3f, Supplementary Fig. 2d, e).

### CDK4/6i-induced modulation of the RB–E2F protein complex is responsible for enhanced viral replication

In order to identify the molecular mechanism responsible for the observed therapeutic effect, we first analyzed the mRNA expression of RB and E2F1 24 h after treatment with PD and adenovirus infection (Fig. 4a, b). Treatment with PD monotherapy showed no significant change in E2F1 and RB transcription at the indicated concentration. Both, E2F1 and RB gene expression levels were increased in a time-dependent manner following Ad infection, which is consistent with the current literature and is further increased by the combination therapy.

We next examined the protein expression of RB, p107, p130, and E2F1 upon treatment with PD and XVir-N-31 or ADWT as mono- and combination therapy in a time kinetic. Downregulation of RB, p107 and E2F1 protein was observed with 500 nM PD monotherapy that remained unchanged over 36 h. No change in the expression level of p130 could be detected. However, in combination with Ad, the initial downregulation of E2F1 could be recovered as early as 12 h post-infection and was fully restored 36 h later, while RB protein was even further suppressed (Fig. 4c). This indicates that Ad-related molecular factors regulate RB and E2F1 protein level even in the presence of CDK4/6i. It also indicates that this newly expressed E2F1 is activated since RB is not present to inhibit E2F in this situation. Considering viral gene transcription, we analyzed the expression of E2-early and E1A13S expression in a time kinetic and observed a robust increase upon PD treatment as early as 8 h post-infection indicating that PD induces expression of viral genes at early time points in the viral life cycle (Fig. 4d, e).

## Expression level of RB but not E2F1 is the decisive factor for viral genome replication

In order to clarify whether the E2Fs affected by CDK4/6i or RB proteins are the crucial factors for the improved viral genome replication, we first downregulated E2F1,−3,−4 and RB1, RBL1 (p107), RBL2 (p130) proteins by siRNA technology (Supplementary Fig. 3a–e). Silencing of the E2F family members 1,3 and 4 that were upregulated by CDK4/6i treatment (Fig. 2e) in T24 cells resulted in a slight increase in viral genome replication, and for E2F4, also viral particle formation without reaching statistical significance (Figs. 5a, b), indicating that these E2Fs are probably involved but not the only decisive factors for the synergistic effect of the combination therapy. Also in the combination with PD, E2F1 silencing did not contribute to a better response to viral genome replication or particle formation (Fig. 5c). In addition, adenovirus infection restores the siRNA-induced suppression of E2F1 24 h past infection but suppresses RB, an effect that correlates with an enhanced expression level of E1A protein 12 h past infection in E2F1 silenced cells (Fig. 5d). This indicates that E2F1 has an important role at later time points in adenovirus replication.

We also transfected SK-N-MC cells that are resistant to treatment with CDK4/6i due to a point mutation in RB[45] with either siRNAs directed against E2F1, RB, p107, or p130. Only with a siRNA against RB, we observed a significant 4-fold increase in viral genome replication (Fig. 5e). This result was confirmed in T24shRB cells that are stable and transduced with an shRNA against RB1 and show significantly reduced RB1 expression level. Upon treatment, only the T24shCtrl cells showed changes in RB level, whereas, in the T24shRB cells, the remaining RB was not significantly altered (Supplementary Fig. 3g) indicating the mitigating effect of the CDK4/6 inhibitor-mediated enhancement of viral replication. ADWT and XVir-N-31 as monotherapy replicated better in T24shRB cells than in the T24shCtrl control cells, suggesting again that RB protein level is an important parameter for regulating viral genome replication. Accordingly, viral genome replication was only slightly improved after CDK4/6i treatment emphasizing the importance of RB1 expression (Fig. 5f).

Consistently, viral genome replication was diminished via restoration of RB in SaOS-2 cells in which the endogenous RB is functionless due to a C-terminal mutation (Fig. 5g)[46]. These results indicate that the expression level of RB protein plays a major role in the initiation of viral genome replication in infected cells.

## E2F binding sites in the adenoviral genome influence viral life cycle

The results obtained with three different CDK4/6i strongly suggest a specific common molecular alteration underlying this positive reinforcement of the adenoviral life cycle. In addition, this assumption is supported by the observation that CDK4/6i treatment facilitated E1A-independent replication and particle formation of dl312 significantly (Fig. 6a, b). This suggests that the treatment of cancer cells with CDK4/6i induces or inhibits a cellular factor (named in the literature E1A-like activity)[23] that not only affects E1A expression (which is mainly responsible for driving viral replication) but could also affect other viral genes, including E2, which is known to be activated by E2F1[43,47].

Plasmid-based promoter analyses have previously been used to identify and study E2F as the main cellular component involved in E2-early promoter activation[48,49]. We cloned the E2F binding sites fused to a luciferase reporter and a mutated version resulting in the pE2-early-luc and pE2-earlyM-luc. As shown in Fig. 6c, the expression of luciferase in the pE2-early-luc was largely suppressed by PD treatment indicating that E2F1 is important for activating luciferase expression. No expression of luciferase could be detected with the mutated pE2-earlyM-luc. Cells were then additionally infected with the E1-deleted adenovirus dl703 or dl348 which do not expresses the E1A12S protein[50]. Infection with dl348 induced luciferase expression also in PD-treated cells whereas infection with dl703 had, as expected, no effect on the E2F regulated luciferase expression. No luciferase expression was observed with the mutant E2F promotor (E2Fm), indicating that E1A expression re-activates E2-expression even after CDK4/6i treatment. These results demonstrate that these constructs respond to cellular factors in the presence of E1A. However, it is important to note that these assays also demonstrate that transcription of E2-early genes in the context of the viral genome are controlled by additional factors since we show in Fig. 4e that PD treatment improves transcription of E2-early.

A precise analysis of the mutated E2F-binding sequence in the E2-early promoter embedded in the viral genome has not yet been performed to date. Therefore, we generated a recombinant adenovirus with these modifications in ADWT (ADWT/E2Fm). We analyzed viral genome replication 24 h post-infection and observed dramatically reduced genome copies of the ADWT/E2Fm compared to ADWT, demonstrating the importance of the E2F-binding sites in the E2-early promoter during viral replication (Fig. 6d).

As an alternative approach to study the role of E2Fs on viral replication, we cloned a cassette containing 20x E2F binding sites of the E2-early promoter in the E3-region and as a control mutated E2F-binding sites (ADWT/Trap and ADWT/TrapM) (Supplementary Fig. 4a, b) that has been described to trap free active E2F[51]. We assumed that this approach would provide us with comprehensive possibilities to analyze the role of E2F binding in the viral life cycle. To ensure the functionality of these newly generated E2F-binding sites, we cloned these cassettes additionally into the luciferase reporter plasmid pE2-early-luc resulting in the pE2-early-luc-Trap and pE2-early-luc-TrapM plasmids and performed transactivation assays.

The combined infection and transfection of cells with ADWT and either pE2-early-luc-Trap or pE2-early-luc-TrapM showed that the additional E2-trap cassette is sufficient to suppress E2-early luciferase expression. As expected, PD treatment had no influence in this construct. However, with the mutated Trap-construct E2F protein complexes are not sequestered so that we observed a significant increase in Luciferase expression. PD treatment also slightly reduced activity of the E2-early luciferase cassette (Fig. 6e). In contrast, trapping of E2F/RB by transfecting cells with the pE2-early-luc-Trap and then infecting with adenovirus is beneficial for viral replication, supporting our findings with siRNAs and CDK4/6 inhibitors. In addition, the increase in adenoviral replication in cells that were transfected with the pE2-early-luc-TrapM treated with PD compared to pE2-early-luc-TrapM control also support the previous data because PD downregulates here the E2F/RB protein level (Fig. 6f).

In accordance with the supportive effect in viral DNA replication in a plasmid-based assay (Fig. 6f), we observed an increased genome replication of ADWT/Trap virus compared to ADWT infected T24 cells, especially at low MOI (Fig. 6g). This effect was most pronounced at 18 h post infection (Fig. 6h). However, these differences diminished with an increasing MOI used for infection, although they were still detectable (Fig. 6g, h). The results obtained so far indicate two aspects for viral genome replication. First, we show that the E2F binding sites in the E2-early promoter in the viral genome (ADWT/E2Fm) are crucial in the viral life cycle and, second that the reduced availability of E2F through binding to additional E2F-binding sites does, surprisingly, not affect the activation of the E2-early promoter and thus viral replication. In contrast, the previous data tend to show that trapping the RB/E2F repressor complex in ADWT/Trap is the reason for the increased genome replication at early time points in the viral life cycle.

## CDK4/6i dramatically increases expression of early genes in ADWT/E2Fm

Next, we compared adenovirus RNA and protein expression of ADWT and ADWT/E2Fm in T24 cells in a detailed 24-h time kinetic. The analyses showed reduced transcription and expression level of E1A and

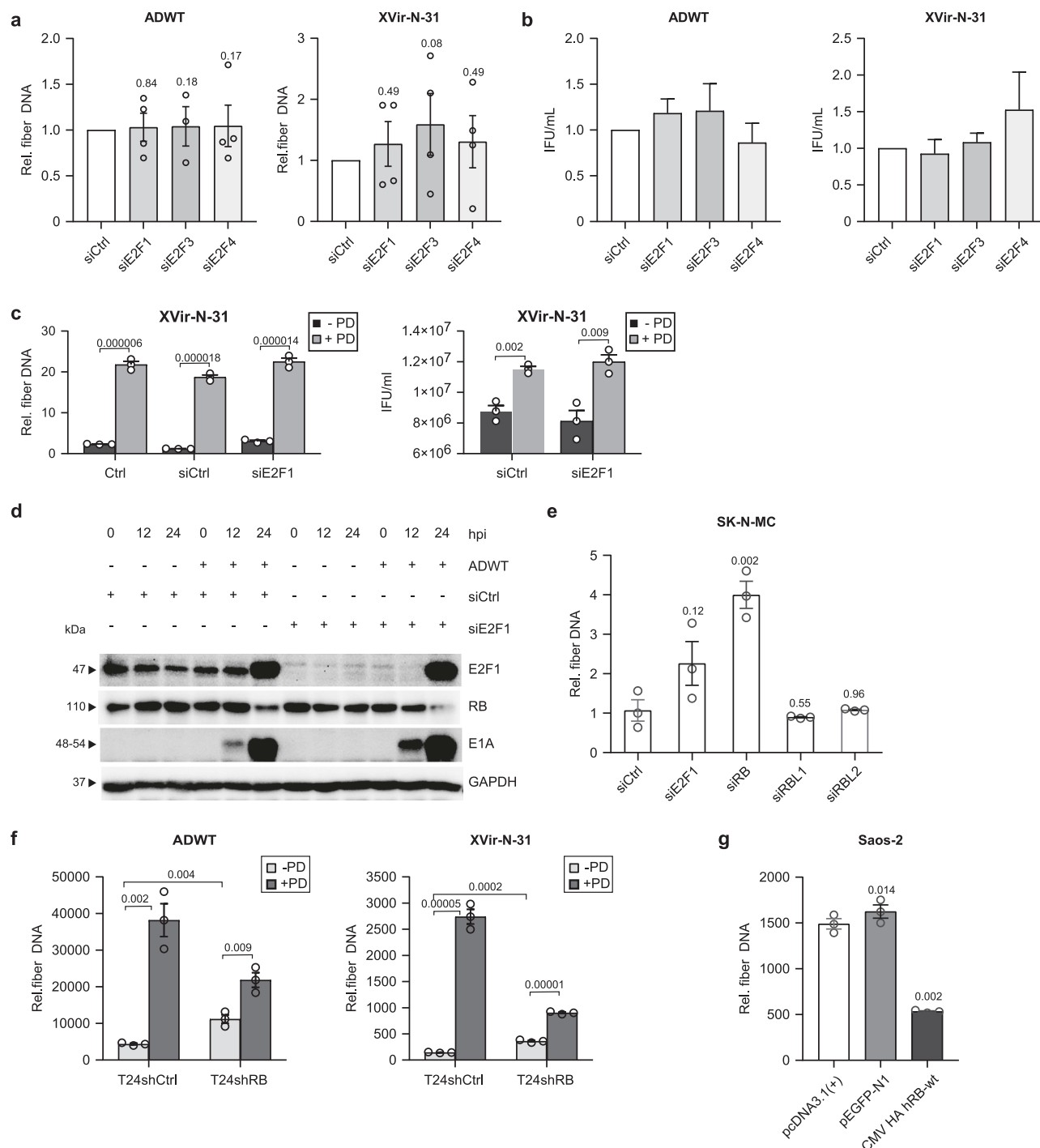

**Fig. 5 | RB but not E2F1 expression level is the decisive factor for viral genome replication. a**, **b** siRNA-mediated knockdown of one of E2F1,3 and 4 was performed with siPOOL technology in T24 cells infected with ADWT or XVir-N-31 (MOI 50). **a** Viral genome replication was analyzed at 24 hpi. *n* = 4. **b** Quantification of viral particles was performed by a hexon titer test and presented as infectious units per milliliter (IFU/ml). *n* = 2, mean ± SE. **c** Viral replication was analyzed in T24 cells transfected with siE2F1 pool and treated with 500 nM Palbociclib for 24 h, and infected with XVir-N-31 (MOI 50). **d** siRNA-mediated knockdown of E2F1 was performed with siPOOL technology in T24 cells and infected with ADWT or XVir-N-31 (MOI 50). Protein levels were detected by immunoblotting at indicated time points for E2F1, RB, and E1A proteins. GAPDH was used as a reference protein. **e** SK-N-MC cells were transfected with siRNAs against E2F1 or RB or RBL1 (p107) or RBL2 (p130)

and infected with the XVir-N-31 (MOI 20). Viral replication was analyzed at 48 hpi. *n* = 3. **f** RB-negative T24shRB1 cells and scrambled control T24shCtrl cells were treated with Palbociclib (1 μM) and infected with MOI 50 of the indicated viruses. Viral genome replication was analyzed at 24 hpi. **g** RB-negative Saos-2 cells were transfected with indicated plasmids and infected with adenovirus ADWT (MOI 20). Viral genome replication was analyzed at 24 hpi. *n* = 3. All viral genome replication analyses were performed by qPCR to amplify viral fiber DNA. Data are represented as relative fiber DNA at indicated time points compared to fiber DNA at 4 hpi as the baseline (mean ± SD). SD standard deviation, SE standard error, MOI multiplicity of infection, hpi hours post-infection. The statistical significance was determined by a two-sided Student's *t*-test; n: number of biologically independent samples. Source data are provided as a Source Data file.

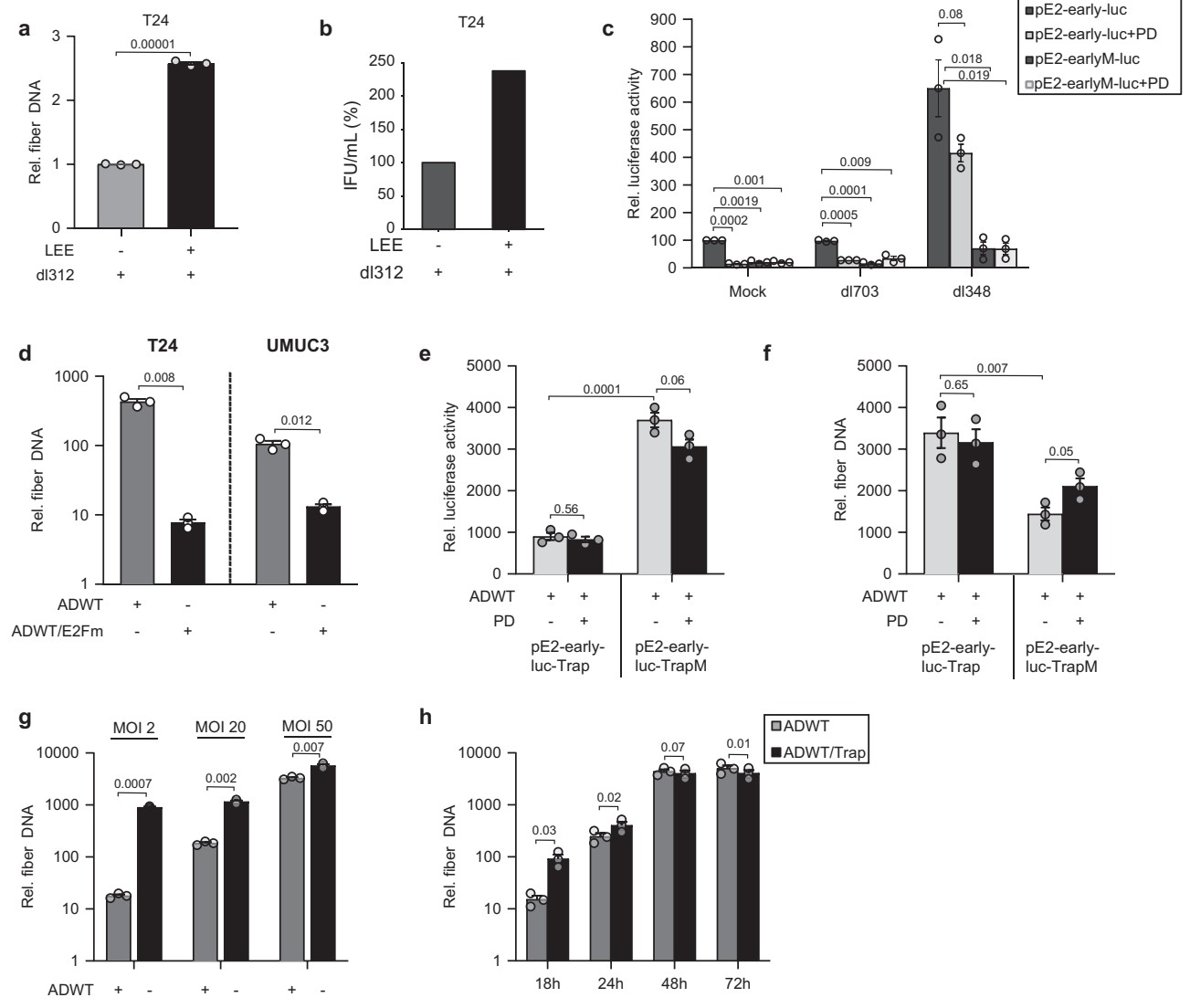

**Fig. 6 | E2F binding sites in the adenoviral genome influence viral life cycle.**
**a**, **b** T24 cells were pre-treated with Ribociclib and infected with the E1A-deleted virus dl312 (MOI 50). **a** Viral genome replication was assessed at 48 hpi. *n* = 3. **b** Quantification of viral particles was performed by a hexon titer test and presented as infectious units per milliliter (IFU/ml) (*n* = 3, mean ± SE). **c** E2-early promoter activity was examined by transfecting T24 cells with plasmids pE2-early-luc or pE2-earlyM-luc. Cells were pre-treated with Palbociclib (0.5 μM) and infected with adenoviruses dl703 or dl348 (MOI 100). Luciferase activity was measured at 42hpi. Results are presented as relative luciferase activity in the percentage of control, (*n* = 3, mean ± SE). **d** Viral genome replication upon E2F-binding site mutation in E2-early promoter was assessed in T24 and UMUC-3 cells that are infected with ADWT or ADWT/E2Fm (MOI 10) at 24 hpi. **e**, **f** The effect of E2F-binding on E2-early promoter activity was assessed in T24 cells that were transfected with a plasmid containing 10 E2-early motifs with 20 E2F binding sites (pE2-early-luc-Trap) or with mutated E2F binding sites (pE2-early-luc-TrapM). Cells were infected with ADWT (MOI 50). Luciferase activity (**e**) (mean ± SE) and viral replication (**f**) were assessed at 24 hpi, *n* = 3. **g** Viral genome replication was analyzed at 24hpi in T24 cells infected with ADWT or ADWT/Trap at indicated MOI, *n* = 3. **h** T24 cells were infected with ADWT or ADWT/Trap (MOI 10) and viral genome replication was analyzed at indicated time points, *n* = 3. All the viral replication analyses were performed by qPCR to amplify viral fiber DNA. Data are represented as relative fiber compared to fiber DNA at 4hpi as baseline (*n* = 3, mean ± SD). SD standard deviation, SE standard error, hpi hours post infection, MOI multiplicity of infection. The statistical significance was determined by two-sided Student's *t*-test; *n* number of biologically independent samples. Source data are provided as a Source Data file.

consequently all other viral genes over time in the ADWT/E2Fm-infected cells (Fig. 7a, b). The effect of replacing the E2F-binding site in the E2-early promoter on E1A expression was initially unexpected. However, it is well established that RB/E2F influences E1A expression by binding to E2F-binding sites in the E1A-enhancer region[52]. Therefore, we speculated that, firstly, by replacement of the E2F-binding sites in the E2-early promoter more RB/E2F was available to bind to the existing E2F-binding sites on the E1A-enhancer and suppress transcription and, secondly, that treatment with CDK4/6i may reverse this effect. Using a ChIP analysis of the E1A enhancer region with either ADWT or the ADWT/E2Fm sequence as bait and an antibody directed against E2F1 for the immunoprecipitation, we observed that the enhancer region of E1A in the ADWT/E2Fm adenovirus showed an increased occupation with E2F1 in the ADWT/E2Fm adenovirus compared to the ADWT (Fig. 7c). As expected, PD treatment which reduces E2F1 level (Fig. 4c) showed less E2F occupation, confirming the applicability of this approach. Based on the results we thought that introducing a mutation of the E2F-binding sites at the E1A-enhancer (ADWT/2xE2Fm) would restore the inhibitory effect on E1A expression of the E2F-binding mutation in the E2-early promoter. As shown in Fig. 7e, while E2-early levels were still downregulated in ADWT/2xE2Fm cells, E1A levels were restored to ADWT level. Thus, the inhibitory

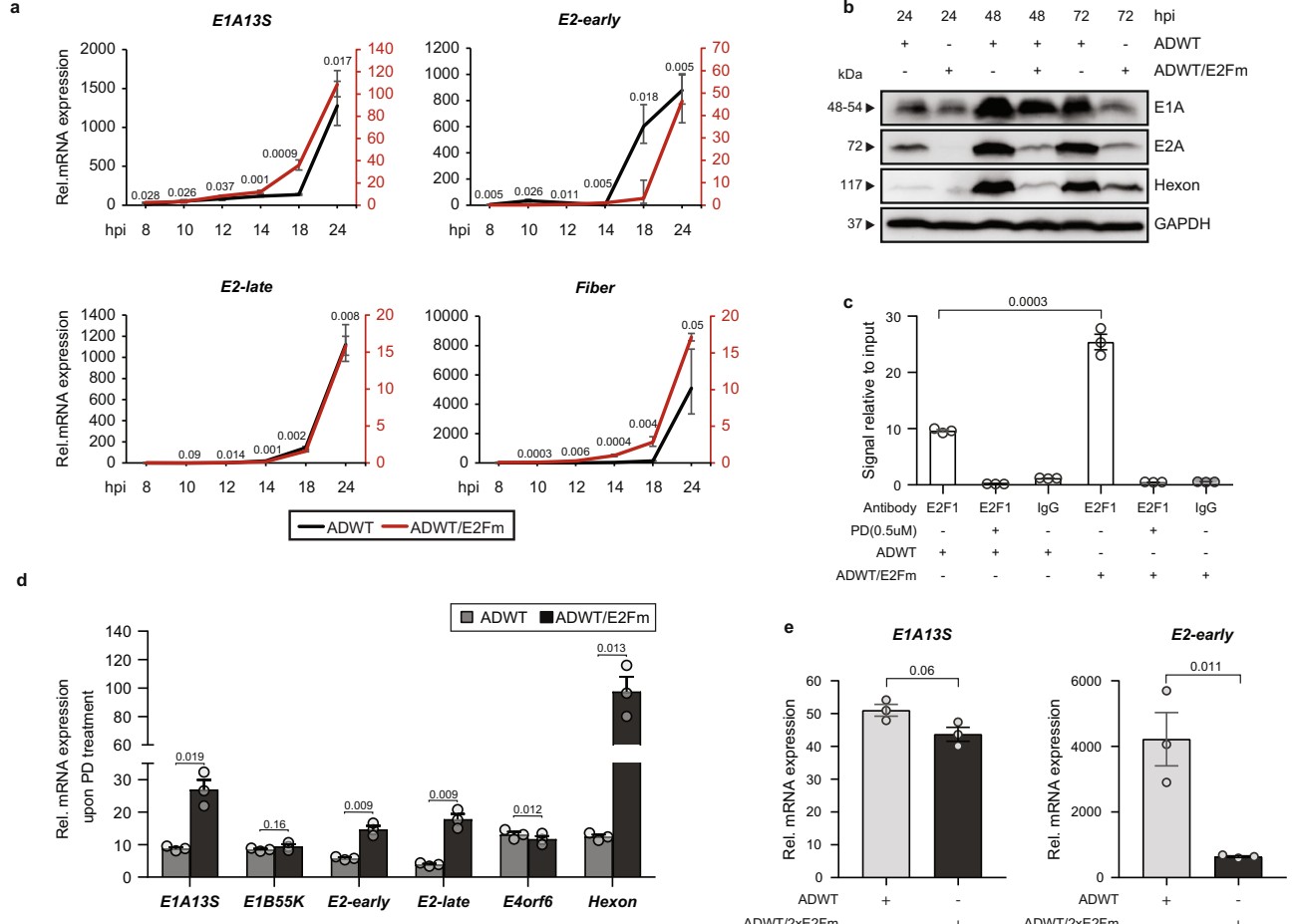

**Fig. 7 | CDK4/6i enhances expression of early genes in ADWT/E2Fm. a** Gene expression was studied in T24 cells infected with viruses ADWT or ADWT/E2Fm (MOI 10) at indicated time points after infection for the viral E1A13S gene, E2-early transcripts, E2-late transcripts and fiber gene. Data are represented as relative mRNA expression compared to the reference gene beta-actin ($n = 3$, mean ± SD). **b** Protein expression was analyzed by western blotting in T24 cells infected with ADWT and ADWT/E2Fm (MOI 10) at indicated time points. One representative blot is shown of three independent experiments. **c** T24 cells, infected with ADWT or ADWT/E2Fm (MOI 50), were crosslinked, isolated and DNA was digested with micrococcal nuclease and subjected to chromatin immunoprecipitation using an E2F1 antibody at 8 hpi. IgG was used as a negative control. Precipitated DNA was quantified by RT-PCR using primers for the E1-enhancer. Palbociclib (0.5 μM)

treated cells were used as the positive control ($n = 3$, mean ± SE). **d** Gene expression for viral genes was analyzed in T24 cells pre-treated with Palbociclib (0.5 μM) and infected with ADWT and ADWT/E2Fm (MOI 50) at 20 hpi. Data are represented as the fold increase relative to untreated control ($n = 3$, mean ± SD). **e** Expression analysis for the viral transcripts E1A13S and E2-early transcripts was performed through qRT-PCR in the cells infected with ADWT or ADWT/2xE2Fm (MOI 10). Data are represented as relative mRNA expression compared to the reference gene beta-actin ($n = 3$, mean ± SD). SD standard deviation, SE standard error, MOI multiplicity of infection, hpi hours post infection. The statistical significance was determined by two-sided Student's *t*-test; n number of biologically independent samples. Source data are provided as a Source Data file.

effect of E1A expression caused by deletion of the two E2F-binding sites in the E2-early promotor is reversed by additional deletion of l E2F-binding at the E1A-enhancer region. These results support the hypothesis that different E2F-binding sites within the viral genome compete for available E2F transcription initiation complexes in order to control and regulate viral replication. It also suggests, that the E2F-binding sites within the E1-enhancer might act as negative regulatory elements.

Since E1A expression was reduced in ADWT/E2Fm-infected cells compared to ADWT thereby affecting the expression of all other viral genes, we hypothesized that treatment with PD would reverse the inhibitory effect of RB to a greater extent than ADWT. As shown in Fig. 7d, PD indeed induced a significantly higher expression of viral genes in both virus constructs but an even higher effect on ADWT/E2FM compared to ADWT.

In summary, our results strongly support the idea that RB is a crucial regulatory element in the adenoviral life cycle and also reveal a hitherto unknown existing cooperation between E1A- and E2-

expression through RB/E2F to ensure a timely coordinated expression of viral genes in the viral life cycle[53].

## Combination of XVir-N-31 and CDK4/6 inhibition is effective in a murine tumor xenograft model

To examine whether biologically relevant advantages of the proposed combination strategy might also translate into therapeutic benefits, we assessed tumor growth of xenografted A673 sarcoma cells in immunocompromised nude mice (Supplementary Fig. 5). We chose this model because T24 bladder cancer cells are difficult to establish as xenografts in mouse models and our data show that this combination therapy is ubiquitous on different tumor entities. Analysis using the open-access web tool TumGrowth indicated that the tumor growth control was significantly increased in animals receiving the combination treatment (combo, LEE + XVir-31) compared to animals treated with monotherapies (LEE or XVir-N-31 only) or control (PBS). The largest differences in treatment response in-between groups were observed at days 12–21 after start of therapy. When performing

longitudinal comparison of tumor growth curves, the combination treatment also showed the strongest antitumor efficacy with a remarkable decrease in tumor volume after the second injection. Of note, viral genome replication was significantly increased in explanted tumors of representative animals receiving the combination compared to XVir-N-31 monotherapy (assessed 2 days after second i.t. injection). In addition, increased expression of E1A and hexon along with decreased RB expression and an increase in caspase activity was observed. This also resulted in a survival advantage for the animals receiving the combination therapy, even in this immuno-compromised xenograft model lacking functional adaptive immunity. Histopathologic evaluation of a central section of the xenograft tumor tissue revealed a higher percentage of tumor cell degeneration and necrosis after combination therapy and XVir-N-31 application compared to application of LEE and PBS application (Fig. 8a–h). To confirm these data, a second Ewing sarcoma cell line in a Rag2−/− γc−/− xenograft mouse model, in which tumor engraftment and growth is faster compared to the nude mouse model, was performed. Here, we also observed a strong increase in viral genome replication, adenoviral E1A and hexon protein level with the combination therapy accompanied by reduced RB levels in vivo (Supplementary Fig. 6).

## Discussion

This study reveals that the combination of CDK4/6i and oncolytic adenovirus improves the oncolytic potency of XVir-N-31. We also demonstrate a mechanistic insight using CDK4/6i and oncolytic adenovirus in this innovative combination therapy approach for the treatment of bladder cancer and sarcoma cells. Treatment with CDK4/6i increase viral gene expression, genome replication, particle formation and significantly enhance the cytotoxicity of XVir-N-31, a YB-1-based oncolytic adenovirus, and ADWT in vitro and in vivo. This is insofar remarkable as CDK4/6i induce cell cycle arrest[54] whereas the currently most accepted model of adenoviral life cycle suggests that successful adenovirus infection requires the induction of S-phase[26]. However, the natural infection of adenovirus occurs in quiescent, non-dividing cells which are most suitable for adenovirus infection and initiation of replication[40]. Nonetheless, since the effect of this combination therapy is also visible in cells that are not primed with CDK4/6i before infection, we can exclude that cell cycle synchronization in G1/G0 is the major underlying molecular mechanism of this combination therapy.

The perhaps most interesting feature of this combination treatment with regard to the clinical translation is the fact that this therapy facilitates adenovirus replication at very low MOIs although the infection efficacy is not altered. This indicates that a certain threshold of viral genomes is essential for the induction of viral genome replication in a normal cellular environment. This threshold and thus the control of repressive or simply not activated molecular components for the transcription of viral genes is dramatically changed upon treatment with CDK4/6i. Consequently, also oncolytic adenoviruses other than XVir-N-31, would benefit from the combination with CDK4/6i[55]. This assumption from the in vitro results was confirmed in murine xenograft experiments. In vivo, only two injections of the oncolytic virus XVir-N-31 in the combination therapy resulted in a significantly smaller tumor size, longer lasting response, higher intratumoral virus titers and consequently a survival benefit of respective animals. The xenograft models used in this manuscript also prove that in a 3-dimensional in vivo situation, the combination therapy used results in a much better oncolytic potency which corresponds to better replication compared to the virus monotherapy. However, for clinical purposes a humanized patient-derived xenograft model would be beneficial since the therapeutic effect of oncolytic viruses strongly depends on activating an anti-cancer immune response in vivo[56].

Mechanistically, our study identified that the inhibition of RB protein by CDK4/6i is a key element of this combination treatment.

This is supported by the observation that therapy response is largely restricted to cells with a functional RB protein. Our data also raise questions concerning the exact role and regulation of E2F proteins throughout the adenoviral life cycle.

We identified RB as a major factor in regulating viral genome replication in this combination therapy. This conclusion is strongly supported by loss and gain of function strategies that were applied in this study. First, the combination therapy is not effective in RB-negative bladder cancer cells and introducing a wild-type RB in SaOS-2 cells[46] that express an RB mutant leads to a reduction of viral replication. Second, downregulation of RB in the CDK4/6-resistant cell line SK-N-MC[57] by siRNA leads to a significant increase in viral replication. The role of RB as a tumor suppressor in cell cycle control is well established, but based on our results RB also serves as a virus suppressor. This aspect needs to be verified by more comprehensive studies, but it has been demonstrated that the E1A–RB complex suppresses the transcription of genes with antiviral functions in adenovirus-infected cells[58].

In order to investigate the influence of the E2F and RB protein family on viral genome replication in context of this combination therapy in detail, we employed several strategies to deplete endogenous E2F and RB proteins. This was achieved by either using siRNA against RB and E2F or by using novel adenovirus vectors that were designed to address this question.

The key role of RB in facilitating adenoviral replication has been identified by the application of reconstitution assays, using plasmid-based RB expression and siRNA technologies against RB. This observation is not surprising since RB acts as a general suppressor of the E2F-family function which is necessary for transcription of the E1A and the E2 promoter. Thus, regulation of pocket proteins obviously controls activity and composition of E2F family members at their appropriate promotor sites in the adenoviral genome. In the case of the combination of CDK4/6i and the oncolytic adenovirus XVir-N-31, our results indicate that RB but not p107 and p130 is the important member of this protein family since its downregulation is sufficient for activating replication. However, the reason why trapping of E2F improves viral replication remains to be studied in deeper analysis. In this context, it is of importance to note that our results show that CDK4/6i monotherapy induce downregulation of both, RB and E2F1. E2F-binding sites have been identified for the E1 and E2 regions in the adenovirus genome[47]. To further examine the role of E2F-binding site containing promotors in the viral genome in greater detail, we replaced the E2F-binding sites in the E2-early promoter (ADWT/E2Fm) which, as expected, causes a reduction in viral genome replication. However, the dramatic reduction in viral gene expression, besides E2, was surprising in particular as we discovered that E1A expression was strongly inhibited. Analyses of this phenomenon revealed a previously unrecognized crosstalk between the E1A-enhancer and the E2-early promoter through the existing E2F-binding sites. This finding is in line with published data postulating such a crosstalk between E1A- and E2 proteins to ensure a coordinated expression of viral genes[59]. In addition, it was reported that IFNγ suppresses Ad replication depending on a conserved E2F-binding site in the E1A-enhancer region but also adenovirus E2 expression probably through enrichment of RB/p107/p130–E2F complexes at the E1A enhancer[60].

One important finding of the described combination therapy with CDK4/6i and XVir-N-31 is the observation that RB expression remains downregulated throughout viral life cycle in the combination therapy whereas E2F1 level recover already 12 h past infection. This observation is important since CDK4/6i monotherapy only temporarily suppresses RB protein level due to cellular mechanisms of acquired resistance, meaning that the continuing suppression of RB is actively mediated by the virus[16,61]. The reason for this unexpected finding is still unclear and remains to be studied further in detail. One explanation might be that RB

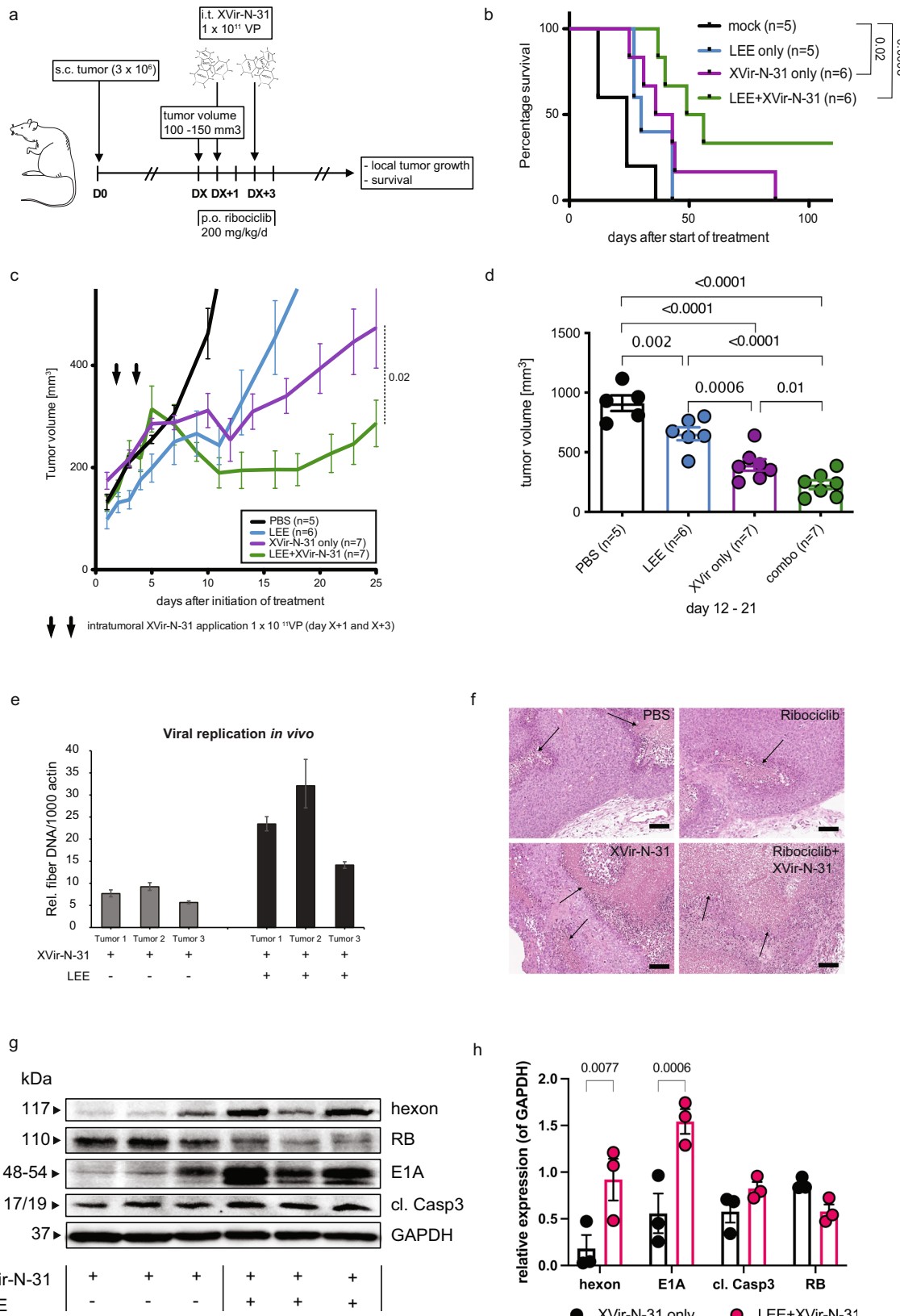

suppression by CDK4/6i does release the activity of remaining E2F1 that initiates in turn expression of E1A. The data presented in the ChIP assay (Fig. 7c) indicate that the E2F-binding sites in E2-early and the E1A enhancer compete for available E2F. Thus, our hypothesis is that CDK4/6 inhibitors, although decreasing the level of both RB and E2F1, result in a pool of free E2F that is first directed to the E1A-enhancer and initiate expression of this essential protein for replication. It has been shown that E1A directly targets the E2F/DP1 complex through interaction with DP1, the heterodimeric binding partner of E2F that might result in stabilization of this RB-depleted complex but would not explain the sustained suppression of RB expression level[62].

**Fig. 8 | Effect of combination and monotherapy in xenograft mouse models.**
**a** Experimental design of the in vivo study (nude mouse model). VP viral particles.
**b** Kaplan–Meier survival curves of treatment groups (*n* = number of animals pre group). Statistics, if not specified otherwise were derived from mixed-effect modeling using the TumGrowth software (see methods section) Survival curves were compared using the log-rank test. **c** Tumor volume growth curves of respective treatment groups. Each data point shows the mean ± SE tumor size at indicated days after initiation of treatment. LEE011 (LEE) was administered daily at 200 mg/kg/body weight by oral gavage for a total of 5 days (day X until day X + 3). All control animals which did not receive XVir-N-31, received intratumoral PBS injections. **d** Evaluation of therapy response at days 12–21 after initiation of treatment is indicated column bars (tumor volume) of respective treatment groups (PBS, LEE, XVir-N-31 only, combo); the number of animals per group is indicated (*n*). **e** Viral

genome replication within the tumor is assessed by qPCR to amplify viral fiber DNA in three independent tumors in monotherapy (XVir-N-31 only) and in combination therapy (LEE + XVir-N-31), mean ± SD. **f** Hematoxylin and eosin staining of the xenografts. Arrows indicate necrotic areas. Scale bars represent 100 μm. **g** A comparable experimental setup as described in (**a**) was used but A673 tumor cells were implanted in Rag2$^{-/-}$γc$^{-/-}$ mice. LEE was also administered daily at 200 mg/kg/body weight for a total of 5 days but XVir-N-31 was only injected once (1 × 10$^{11}$ VP) at DX + 2 and whole tumors were harvested at DX + 5 for immunoblotting analysis. Each lane presents one explanted tumor and **h** depicts the densitometric quantification of the immunoblots per group and protein, as indicated (done with ImageJ 1.53k). Tukey's multiple comparison in combination with two-way ANOVA was used for statistical analysis (GraphPad Prism). Source data are provided as a Source Data file.

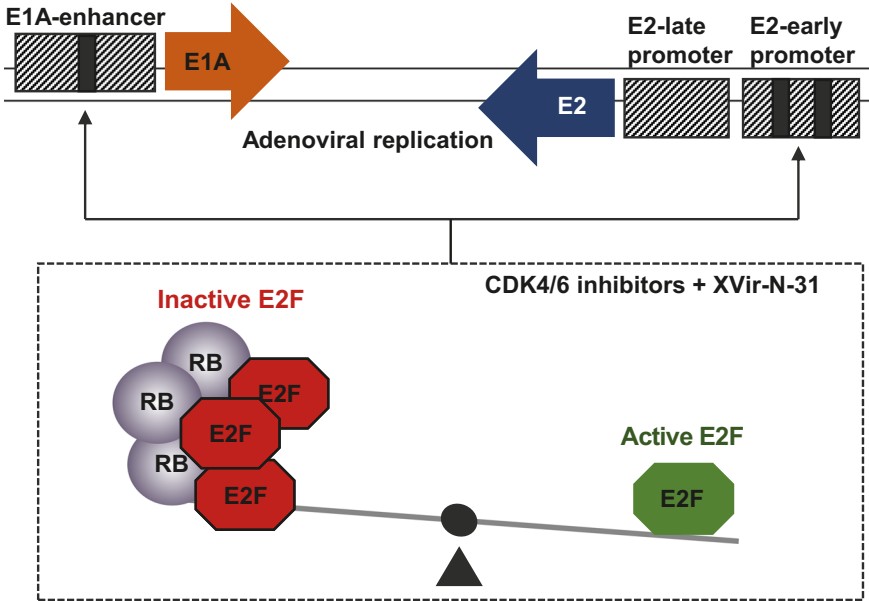

**Fig. 9 | Control of viral replication in the combination therapy with oncolytic adenovirus and CDK4/6 inhibitors.** Adenovirus E1A and E2 expression is controlled by the well-balanced E2F/RB protein complex through E2F-binding sites in their promoter region (black bars). Both promoter regions compete for binding of the E2F/RB protein complex indicating a novel level of regulatory mechanism in

viral life cycle. Excess level of this protein complex suppresses efficient transcription at early time points in the replication life cycle. Inhibition of RB and E2F1 protein level by CDK4/6-inhibitors might cause a shift towards low level of active E2F1 and thus facilitate early viral gene expression and replication.

The data above do not explain to full extent as to why the ADWT/Trap replicates better than the ADWT, since, besides RB, also E2F is sequestered from its ability to activate viral gene expression. One explanation might be the additional tight control of E2F activity by viral proteins during the viral life cycle, indicating that E2F proteins are only active in the context of molecular co-factors, such as E4orf 6/7[63]. Another possible explanation is the formation of nuclear sub-compartments, described as viral inclusion bodies/adenovirus replication compartments[64]. These compartments provide an optimal environment for virus replication by concentration of viral and cellular factors beneficial for the virus, while excluding inhibitory factors. As an example, the translocation of YB-1 into viral inclusion bodies facilitates viral genome replication, while its exclusion from the cytoplasm impairs cellular protein synthesis due to its participation in translational regulation[32,65]. However, trapping of E2F raises novel questions on the role of E2Fs in viral replication that might reveal new directions in the mechanism of adenoviral replication. Our studies using a trapping strategy for E2F–RB protein complexes and the RB inhibition by CDK4/6i indicate that RB protein is the essential molecular component as a shift in the well-balanced free E2F and complexed E2F (E2F/RB) system towards active E2F favors viral genome replication.

In summary, we identified a combination therapy using the oncolytic adenovirus XVir-N-31 and CDK4/6i. Our data on the molecular mechanism of this combination strongly support the idea that the RB/E2F complex acts as a repressor in the adenoviral life cycle as depicted in Fig. 9. Moreover, our results reveal a hitherto unknown existing cooperation between E1A- and E2-expression through the RB/E2F complex that ensures a timely coordinated expression of viral genes throughout the viral life cycle. Finally, our discovery provides additional potential therapeutic strategies to enhance adenoviral replication. It is well established that the RB/E2F and p53/MDM2 pathways interact and by that influences E2F activity. Published data have shown that treatment with Nutlin-3a, a MDM2 antagonist, also negatively effects RB expression[66], indicating the usefulness of also targeting this pathway to enhance viral genome replication alone or even in combination with CDK4/6i[67].

Recently, it has been discovered that the overall anti-tumor activity of CDK4/6i might not only rely on their direct anti-proliferative effects on cancer cells, but also on initiating an anti-tumor immune response[68,69]. CDK4/6i also induce a metabolic modulation that promotes chemokine-mediated recruitment of T-cells into mammary tumors[70]. If this assumption is clinically true (ClinicalTrials.gov Identifier: NCT02778685), the capacity of XVir-N-31 (which induces a

stronger immunogenic cell death as ADWT) and other oncolytic Ad to kill cancer cells and by that initiate an anti-tumor immunity provides convincing rationale to further combine this approach with immunotherapies such as immune checkpoint inhibitors[35,66]. These data highlight not only the potential utility of CDK4/6 inhibitors to overcome immunotherapy resistance but also strongly suggest the use of XVir-N-31 as a second-hit immunotherapy agent in patients.

## Methods

### Cell lines

T24 (HTB-4™) and Hek293 (CRL-1573), A673 (CRL-1598) cells were purchased from ATCC, VA, USA. SK-N-MC (ACC 203) and RT112 (ACC 418) cells were obtained from the German Collection of Microorganisms and Cell Cultures (DSMZ, Germany). TC32 was obtained from the Childhood Cancer Repository (CCR, Alex's Lemonade Stand Foundation, Children's Oncology Group, COG). UMUC-3, 639V and 647V were obtained from Professor WA Schulz, Düsseldorf, Germany. 253J cells were a kind gift from Professor G. Unteregger, Homburg, Germany. Authentication of cell lines was confirmed by genotyping analyzing short-tandem repeats in 2016. Cells were cultured under subconfluent conditions either in RPMI supplemented with 10% fetal bovine serum (FBS), 1% NEAA penicillin–streptomycin or Dulbecco's modified Eagle's medium (all from Biochrom GmbH) supplemented with 10% fetal bovine serum and penicillin–streptomycin, at 5% or 10% $CO_2$, respectively. The generation of T24shCtrl and T24shRB cells has been described previously[71].

### Small molecule inhibitors

UCN-01 (Sigma-Aldrich, #U6508) and AZD7762 (Selleck Chemicals #S1532) were dissolved in DMSO as 1 and 10 mM stock solutions, respectively. PD-0332991 isethionate (Palbociclib, MedChem Express, # HY-A0065/CS-3110) and LY-2835219 (Abemaciclib, MedChem express, #HY-16297A) were dissolved in sterile water as 10 mM stock solutions. LEE011 (Ribociclib, MedChem Express, #HY-15777) was dissolved in DMSO as a 10 mM stock solution. Working concentrations were freshly prepared in culture media for immediate use.

### Plasmids and siRNA transfection

Cells were transfected with plasmids pEGFP-N1 (Clontech, GenBank Accession #U55762), pCMV HA hRB-wt (Addgene, #58905), and pcDNA3.1(+) (Invitrogen, V790-20) using FuGENE® HD (Promega, #E2312) transfection reagent according to the manufacturer's protocol. For siRNA-mediated knockdowns, cells were transfected with siE2F1 (Sigma-Aldrich, SASI_Hs01_00162220) siRB (Qiagen, SI00007091), or scrambled siRNA (Qiagen, #1022076) using Lipofectamine RNAiMAX (Thermo Scientific, #13778) according to manufacturer's instructions with a final siRNA concentration of 25 pmol. For E2F1,3 and 4 siPOOL (siTOOLS Biotech, Planegg, Germany), transfection was performed following the siTOOLS Biotech transfection protocol with a final siPOOL concentration of 1 nM. For both plasmid and siRNA transfections, inhibitors were added 6–8 h post-transfection, and infections were performed.

### Adenoviruses and infection

We used dl309 (designated as ADWT in the manuscript) as control[72]. The mutant adenovirus dl312[73] does not express the E1A proteins due to a deletion of base pairs 448–1349. The mutant adenovirus dl348 does not express the E1A243R protein[52] and was a gift from Dr. Alemany (L'Hospitalet de Llobregat, Spain). dl703 contains extended deletions in early region 1[74]. The oncolytic adenovirus XVir-N-31 was described recently[35]. For the construction of ADWT/E2Fm, the E2-early promoter-bearing region between restriction sites Sse232I (nt26390) and EcoRI (nt27326) was exchanged for a synthesized fragment, in which the two E2F-binding sites in the region from nt27082 to nt27113 were changed from 5'-<u>TTCGCGCCC</u>TTTCTCAAATTTAA<u>GCGCGAAAA</u>

-3' to 5'-<u>ACCTATTAC</u>TTTCTCAAATTTAA<u>ATCCGACTC</u>-3' (wild type and mutated E2F-binding sequences are underlined). For the construction of ADWT/Trap, ten repeats of the E2 early promoter sequence segment 5'- TAAGGACTAGT<u>TTCGCGCCC</u>TTTCTCAAATT-TAA<u>GCGCGAAAA</u>CTACGTCAT-3' corresponding to nt27071 to nt27122 (E2F-binding sequences are underlined) were inserted into the two DraI sites in the E3 region at position (nt28706–nt29308). This deleted the E3gp19K and inserted 20 E2F-binding sites. For the production of the double mutant adenovirus variant (ADWT/2xE2Fm), the proximal E2F-binding site in the E1A enhancer was mutated by replacing CGCG with AAAT at position nt 278–281 and both E2F-binding sites in the E2 early promoter were mutated as described for ADWT/E2Fm. The variant is E3 deleted and has an RGD on the HI loop of fiber. The information on the nucleotide positions refers to the NCBI GenBank entry AY339865.1. All adenoviruses were produced in Hek293 cells and purified by two consecutive CsCl-gradient centrifugations and additionally by size-exclusion chromatography using disposable PD-10 desalting columns (GE Healthcare).

For infection, cells were seeded in 10 cm, 6- or 12-well plates, depending on the assay, and treated overnight with desired concentrations of inhibitors. 24 h after inhibitor treatment, cells were infected with viruses in the desired MOIs in 2 mL, 400, or 250 μL serum-free medium for 1 h, unless otherwise stated. Growth medium (2 or 1 mL) was then added to the cultures with and without inhibitor. Cells were processed for further analysis at specific time points after virus infection.

### Cytotoxicity assay

To determine cell viability after treatment with small molecule inhibitors, 500–700 cells were seeded and grown overnight in a 96-well plate. Triplicates of cells were treated with increasing concentrations of the inhibitors for 72 h. Controls were treated with DMSO or water. Cell viability was measured using CellTiter-Blue assay (CellTiter-Blue® Cell Viability Assay, Promega, #G808B) according to the manufacturer's instructions. To determine virus-induced cell killing, cells were seeded in 12-well plates and infected as previously described. Four days post-infection, cells were fixed with 10% cold trichloroacetic acid at 4 °C overnight and stained with 0.5% sulforhodamine B (SRB, Sigma-Aldrich, #S9012) in 1% acetic acid for 30 min. To quantify the cytopathic effect, the bound SRB was dissolved with 10 mM Tris buffer (pH = 10). The absorbance was then measured at 590 nm. The combination index was calculated according to the Chou–Talalay method[75].

### Viral replication and particle determination

$5 \times 10^4$ cells were seeded in six-well plates and grown overnight, treated with inhibitors, and infected as described above. At desired time points, cells were harvested, and DNA was extracted using the Phenol–Chloroform–Isoamylalcohol (Sigma-Aldrich, #P3803) method. Viral replication was analyzed by real-time qPCR using Bio-Rad CFX96 Touch Real-time PCR System to amplify viral fiber DNA using the ΔΔCT-method. Actin was used as the reference for the content of cellular DNA[35] (Supplementary Table 1). Quantification of infectious viral particle formation was measured by immunocytochemistry staining of comparably infected Hek293 cells seeded in 24-well plates using a goat-anti-hexon antibody (Merck, #AB1056), a HRP-conjugated rabbit-anti-goat antibody (Dako, #P0449), and DAB solution (Dako, #K3468). Cells were counted randomly in 10 fields across the wells and the viral titer is calculated as Titer (IFU/ml) = ( average number of positive cells/field * fields/well)/(volume of diluted virus used per well (ml)*dilution factor). Hexon staining in infected cells was performed in a methodologically comparable manner.

### Gene expression analysis

For RNA quantification, $0.5–1 \times 10^5$ cells were seeded in a six-well plate and infected as described above. RNA extraction (phenolic

acid:chloroform, Ambion, #AM9720), reverse transcription (High-Capacity cDNA reverse transcription kit, Thermo Fisher, #4368814), and qPCR (GoTaq qPCR Master mix, Promega, #A6010) was performed according to the manufacturers' instructions. 1 µg RNA was used for reverse transcription. For viral genes, gene-specific reverse transcription was performed with a primer concentration of 0.1 µM. The applied primers are described in the Supplementary information as Table 1. The cDNA was analyzed by real-time PCR using the Bio-Rad CFX96 Touch Real-time PCR System. Relative quantification was performed using the ΔΔCT-method. Samples were normalized to actin cDNA levels and/or untreated controls. The primer sequences are listed in Supplementary Table 2.

## Immunoblot analysis

Approximately $2 \times 10^6$ cells were lysed on ice in a buffer containing 1% SDS, 1 mM sodium orthovanadate, 10 mM Tris (pH 7.2), and protease inhibitor cocktail (Roche Diagnostics, #05892970001). Lysates were sheared with a 27-gauge needle (BD Biosciences, #305109) until no viscosity was observed and then centrifuged at $30,000 \times g$ at 4 °C. Protein concentrations were quantified using Pierce BCA Protein Assay (Thermo Scientific, #23225). Proteins were loaded on 10–12% polyacrylamide gels for SDS−PAGE and blotted on PVDF membranes (GE-Healthcare, #10600021). Subsequent immunoblotting was performed as described previously[16]. Antibodies used and conditions are listed in Supplementary Table 3.

## Immunofluorescence

$1 \times 10^4$ cells were grown directly on coverslips and treated with the inhibitors. 24 h after treatment, cells were fixed using ice-cold methanol-acetone (1:1) for 15 min at −20 °C. Fixed cells were blocked with 3% BSA (in PBS) and stained for YB-1 (Abcam, #EP2706Y) followed by incubation with secondary anti-rabbit Alexa Fluor 488-conjugated antibody (Invitrogen, #A11008). Slides were mounted with ProLong® Gold Antifade Mountant with DAPI (Thermo Fisher, #P36931). Images were taken with the AxioVert.A1 Microscope and Camera AxioCam ERc5s (Zeiss) or an Evos M5000 from Invitrogen.

## Electron microscopy

For ultrastructural analysis, confluent monolayers of infected cells were washed three times with 0.1 M sodium phosphate buffer (pH = 7.2) and fixed with 2.5% glutaraldehyde in the same buffer for 30 min at room temperature. The cell layers were washed again with buffer and fixed with 1% osmium tetroxide for 30 min. For further processing, the fixed cells were scraped from the culture dishes, collected by centrifugation, embedded in low-melting agarose, and subsequently dehydrated, infiltrated, and embedded in EPON™ resin according to standard procedures. Finally, thin sections were cut from resin blocks, mounted on 200-mesh copper grids, and stained with uranyl acetate and lead citrate. The sections were examined with a Zeiss EM10CR transmission electron microscope at 60 kV.

## ChIP analysis

For ChIP analysis, cells were grown to 80% confluency in 10 cm dishes and infected with ADWT-RGD or ADWT/E2Fm at MOI 50 and harvested 8 h after infection. Cells primed with 500 nM Palbociclib before infection with ADWT-RGD served as the negative control. ChIP was performed according to the Pierce™ Agarose ChIP Kit protocol (Thermo Scientific, #26156) with the modification that RNase digestion was extended to 8 h. Immunoprecipitation was performed with monoclonal E2F1 antibody (Cell Signaling, #3742). The immunoprecipitated DNA was quantified by qPCR to amplify the E1A-enhancer region. The primer sequences are listed in Supplementary Table 2.

## Luciferase transactivation assay

Cells were seeded in six-well plates, allowed to grow overnight, and were transfected with pGL3 luciferase reporter constructs (E2-early promoter corresponding to nt27194–nt26981 from Human adenovirus C serotype 5, GenBank: AY339865.1) as wild type or with mutated E2F-binding sites. 300 ng of plasmid was transfected into the cells using FuGENE® HD transfection reagent (Promega, #E2311). After transfection, 500 nM of Palbociclib was added and 8 h later cells were infected with different adenoviruses. 42 h post-infection, cells were washed with ice-cold PBS and lysed by adding 400 µl of cell culture lysis reagent (Promega, #E153A). Lysates were centrifuged at $6000 \times g$ for 5 min. The luciferase activity in the supernatant was measured by a Victor X3 Plate Reader (Perkin Elmer) after mixing 20 µl of the supernatant with 100 µl of luciferase assay substrate (Promega, #E151A).

## Caspase 3/7 and cell death assays

To determine caspase-dependent apoptosis after treatment with Palbociclib, 2000 cells per well were seeded and grown overnight in 96-well plates. Triplicates were treated with increasing concentrations of the inhibitors for indicated amount of time. Caspase-Glo 3/7 (Promega, #G8091) and Cell Titer-Blue (Promega, #G8081) assays were conducted in parallel according to the manufacturer's instructions. The caspase 3/7 activity from the Caspase-Glo assay was normalized to the number of cells present in each condition as determined by Cell Titer-Blue assay. In parallel, $2 \times 10^6$ were seeded in 10-cm plates overnight and treated with increasing concentrations of the inhibitors. Medium was aspirated and cells were harvested and stained with Trypan Blue. Stained cells were visualized and quantified using Invitrogen EVOS M5000 microscope and counting ×10 optical fields.

## Mouse studies

Female NMRI-Foxn1 nu/nu mice (nude mice) were purchased from Charles River Laboratories (import stock #5180) and Rag2$^{-/-}$γc$^{-/-}$ mice (BALB/c background, initially obtained from the Central Instiue of Experimental Animals, Kawasaki, Japan; re-imported from Charles River Laboratories after embryonal transfer) were bred and maintained in our animal facility (Center for Preclinical Research, School of Medicine, TUM) under SPF conditions in consideration of dark/light cycle, ambient temperature and humidity. Experimental and control animals were bred/co-housed within the same room of our animal facility. Animal experiments were approved by local regulatory authorities according to German Federal Law and in accordance with institution guidelines (permission numbers of Regierung von Oberbayern: 55.2-2532.Vet_02-17-225, 55.2-2532.Vet_02-15-102, 55.2-2532.Vet_02-20-165). Experiments were performed on 10–20-week-old mice. For the human sarcoma xenograft nude mouse model, mice were injected subcutaneously (s.c.) in the right flank with $3 \times 10^6$ A673 tumor cells in PBS. Tumor size was measured every 2–3 days with a caliper and tumor volume was calculated with the formula volume = $0.5 \times$ length $\times$ width$^2$. After the tumor volume exceeded 100–150 mm$^3$, mice were randomized into indicated treatment/control groups: PBS (i.e. 0.5% methylcellulose without LEE011 and PBS intratumorally [i.t.]), LEE (i.e. 0.5% methylcellulose containing LEE011 and PBS i.t.), XVir-N-31 only (i.e. 0.5% methylcellulose without LEE011 and XVir-N-31 i.t.) and combination (i.e. 0.5% methylcellulose containing LEE011 and XVir-N-31 i.t.). Then, respective animals received Ribociclib succinate (LEE011) 200 mg/kg body weight (dissolved in 0.5% methylcellulose) on day X (DX) until day X + 4 (DX + 4) or mock control (0.5% methylcellulose without LEE011) via oral gavage. $1 \times 10^{11}$ viral particles (VP) of XVir-N-31 or PBS (in 50 µL respectively) were injected i.t. on day X + 1 (DX + 1) and day X + 3 (DX + 3). On day X (start of therapy) +5, three representative animals from the "XVir-N-31 only" and "combination" treatment groups were euthanzied for histopathological assessment and quantification of viral replication in explanted tumors. The tumor size of the remaining mice was measured until the tumor volume reached

1000 mm³ which was the permitted maximal tumor burden from our regulatory authority. The day the tumor volume reached or exceeded 1000 mm³ or mice reached other end-point criteria (i.e. ulcerating tumor), mice were euthanized using isoflurane and cervical dislocation. No significant weight loss was stated between treatment- and control groups and the all end-point cirteria were respected in this study. Confirmatory experiments with regard to viral genome replication, viral and cellular protein expression were performed in female 10–20-week-old mice Rag2$^{-/-}\gamma$c$^{-/-}$ mice using A673 and TC32 cells for s.c. tumor inoculation. LEE was administered daily at 200 mg/kg/ body weight for a total of 5 days but XVir-N-31 was only injected once ($1\times10^{11}$ VP in A673 xenogtafts, and $5\times10^{10}$ VP in TC32 xenografts) at DX + 2 and whole tumors were harvested at DX + 5 for further analyses. Numbers of animals are given in respective figures.

### Histopathology
Tissue was fixed in 3.5% paraformaldehyde for 24–72 h at 4 °C, followed by dehydration in a tissue processor and embedded in paraffin. 3.5 μm-thick sections were cut using a microtome (Leica) and placed onto slides that were used for hematoxylin & eosin (H&E) staining.

### Statistical analysis
For the in vitro experiments, the data are expressed as the mean ± SD/ SE, and the comparisons were performed using two-tailed Student $t$ tests. Tumor growth in vivo was analyzed with the open-access web tool TumGrowth[76]. Briefly, measured data of tumor volumes were subjected to linear mixed-effect modeling, allowing longitudinal tumor growth slope comparison as well as treatment response evaluation at desired points in time (cross-sectional analysis). $P$-values were calculated by the software, using type II ANOVA and selected pairwise comparison for longitudinal and cross-sectional analyses (with holm adjustment, when indicated). Tumor growth curves and Kaplan–Meier curves were generated using Prism 5 (GraphPad Software). In the tumor growth curves, the mean tumor volumes and standard errors of the means (SEM) are indicated. $P$-values < 0.05 were considered statistically significant.

### Reporting summary
Further information on research design is available in the Nature Research Reporting Summary linked to this article.

## Data availability
Source data are provided as a source data file. Source data are provided with this paper.

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

## Acknowledgements

This research was supported by the Deutsche Forschungsgemeinschaft (DFG), project numbers NA439/2-1 and HO 1482/7-1.

## Author contributions

Conceptualization and supervision: P.S.H. and R.N.; Funding acquisition: P.S.H. and R.N.; Investigation: J.K., S.J.S., S.V.H., C.S., F.G.K., K.M., M.E., U.S., T.H., P.Q., K.S., M.A.; Writing—original draft, review, and editing: R.N., J.E.G. and P.S.H.; All authors have read and agreed to this version of the manuscript.

## Funding

## Competing interests

P.S.H. is co-founder of XVir Therapeutics GmbH. All other authors declare no competing interests.
