## [Peer Review File · Nature Communications]

Targeting the Retinoblastoma/E2F repressive complex by CDK4/6 inhibitors amplifies oncolytic potency of an oncolytic adenovirusReviewers' comments:

Reviewer #1 (Remarks to the Author):

In the manuscript by Koch et al the authors attempt to show that inhibition of cdk4 and 6 kinases, which play a critical role in the G0 to G1 transition, enhanced adenovirus growth and can be combined with an oncolytic virus to enhance tumor cell killing. The authors proceed to show that treatment of a variety of cancer cell lines with these inhibitors enhances virus growth, including that of the wild-type adenovirus. The authors attempt to elucidate the mechanism of the perceived enhancement by investigating a potential molecular mechanism that relies on the pRb protein rather than E2F1, and this may not be E1A-dependent as enhancement with a cdk inhibitor also enhanced dl312 growth, which is a mutant that does not express E1A. Overall the paper is very lengthy but ultimately the findings are unconvincing. Importantly, the authors seem to lack a complete understanding of how adenovirus affects the E2F-pRb axis, and cell cycle regulation overall. Furthermore, the authors completely ignore the important role of E2F1 in apoptosis, which may play a critical role in some of their findings, and importantly the authors misunderstand many aspects of virology, for example, by equating viral genome replication with virus replication. The authors also must show how E1A occupancy on E2F and viral promoters is affected by these inhibitors, and whether other E2Fs play a role. The manuscript itself is written well enough, with some minor language issues and a few minor typos. Ultimately however, the findings are preliminary and the conclusions are not fully supported by the data. Importantly, the mechanistic insights into what is happening are largely missing or grossly oversimplified, lacking much of the detail necessary to completely understand what is happening. As such, I do not recommend publication of this manuscript. Extensive work needs to be done to resolve many serious issues and omissions currently present. Below, I outline all of my major and minor points in the order of appearance within the manuscript (mostly).

Major:

1. Line 64. What about contribution of free E2F1 to apoptosis that would be expected under inactive/degraded Rb condition? This should be discussed in the intro, and is likely contributing to some of the observed effects especially with wt virus, but likely the mutant as well. What about 01/PEME virus published by Ramachandra that relied on defects in these pathways to drive selectivity, a virus that came nearly 20 years ago with much higher and complex cancer specificity? Enhanced E2F activity was shown by Howe et al (2006) with a wt E1A cell line leading to early cell death and reduced virus growth. This was mitigated by E1A 01/07 deletion that abrogates p300 and Rb binding. These should be discussed in the introduction.
2. Fig 1c. Is there an infection here? If showing selectivity here is the objective, then the authors need to show that ADWT can replicate. Furthermore, the authors should show that these cells are getting infected properly and that they're expressing at least E1A if not the other viral genes.
3. Fig 1e and numerous others. Assaying for viral genomes is not the same as for viral particles, therefore this is not accurate. For example, most rodent cells will replicate human adenovirus DNA but no infectious particles will be made so quantifying vDNA replication and equating it to viral replication is wholly incorrect. The authors should show actual viral replication from viruses not genomes extracted from infected cells. Whether this is done thru PCR or plaque assay doesn't matter, but equating viral genomes in infected cells to viral replication is wrong.
4. Fig 1d. Why is the level of Chk1 vastly different at no drug (0 concentration) between UCN-01 and AZD7762 treated cells? At time 0 there should be no treatment and no difference.
5. Fig 2a. Why is there no replication of ADWT with no drug treatment? This virus is expected to replicate regardless, but here it is shown not to or very little. This makes no sense whatsoever. Therefore, a control showing that these cells are infected and producing viral proteins is needed. Also, the authors need to show how long it takes the cells to be killed by the infection as compared to treated cells, which seem to die by day 4? Regardless, the WT virus should always kill the cells, so I do not understand why it is not.
6. Fig 2g. ADWT should be shown in these cells as well to indicate that there is some degree of selectivity. Furthermore, if the XVir-N-31 is selective oncolytic virus, it needs to be shown that it does not kill normal cells with cdk4/6 inhibition, which is not shown anywhere in the manuscript.
7. Fig 2f. Why is there only 1 E1A band in the ADWT lanes? E1A is known to produce 5 isoforms, 4 of which are easily detectable during infection by western.

8. Fig 2f and a. Hexon production also indicated that viruses are being assembled, particularly with ADWT, yet cell viability is little compromised in (a). What is the reason behind this as it doesn't add up? Generally, once hexon production is initiated the obvious CPE is visible, was cellular morphology altered in (a)?
9. Line 148. What is the level of CAR expression on the different cell lines? The authors need to show this either by western, or qRT-PCR at the least.
10. Fig 3d. The observation of lack of enhancement of infection/replication in 293 cells versus the others suggests that this may entirely be an E2F1-apoptosis driven mechanism. Inhibition of cdk5 would preclude E2F release from pRb, and prevent activation of pro-apoptotic genes by free E2F1, in 293 cells, that constitutively express E1A, this would have little effect as Rb is already inactivated by E1A and further inactivation would have no to little effect. This possibility was not investigated by the authors whatsoever and it needs to be, it is a critical aspect of the potential mechanism.
11. Fig 4c. What are the levels of the other E2Fs and other pocket proteins (p107 and p130), particularly p130 which can often substitute of pRb under certain pRb-deficient conditions.
12. Fig 4d. Why is there a drop of E1A/E2 levels? This should go up continuously, especially E2 as was previously shown by Crisostomo et al 2019. If this is due to the way the data is presented, I think a better and more informative way would be to present this relative to a housekeeping gene rather than fold change versus wt.
13. Fig 5. Blots of knockdown and/or overexpression must be shown for all genes.
14. Fig 5c. Is the reduced genome replication due to inability of E1A to be recruited to pRb-repressed promoters in the presumed absence of pRb?
15. Line 212-213, supplementary figure. This is insufficient as changing sequence is not likely to affect mRNA expression but might affect protein sequence/stability, etc. Therefore, the authors need to show that viral proteins (L4-encoded) are not compromised in expression by the introduction of these mutations.
16. Line 214. Genome replication, not virus. Also, was the level of E2 proteins affected by this mutant as compared to wt? This needs to be shown.
17. Line 219-220. Assuming that E2F1 is the only E2F that will bind to these sites is grossly simplistic, furthermore, it totally ignores the important role of E1A in transactivation of ALL viral promoters.
18. Fig 7c. This experiment is missing the important IgG negative control and the data should be represented as % of input rather than copy number.
19. Fig 8. Where is the ADWT in these studies?
20. Discussion. What happens when the Rb-binding site of E1A is deleted?
21. Discussion. I suspect that inhibition of cdk4/6 results in enhanced complex of E2F/pRb, which allows greater E1A access to certain promoters via pRb, or perhaps directly through E2F. This should be determined. The authors need to show whether inhibition of cdk4/6 stabilizes the E2F/pRb complex and whether E1A is present in this complex during infection, and whether it accesses promoters through this interaction. The authors also fail to look at other E2Fs, particularly repressive ones that E1A is known to bind to preferentially and directly (via heterodimerized DP-1).
22. Line 319. This is consistent with E1A having more access to E2Fs in the absence of pRb.
23. Line 334. Authors claim that only pRb is important is not based on evidence, as the manuscript never investigated the role of p107 nor p130.
24. Discussion. There is no discussion or investigation of apoptosis driven by high E2F1 activity. This needs investigation.
25. Statistics are largely missing from most of the data figures.
26. The role of E4orf6/7 in nuclear translocation of E2F4 is not mentioned or investigated. Since E4 is driven by E1A, E1A lacking CR3 will drive this gene poorly, leading to lower levels of E4 proteins (at least early in infection), which could affect E2F4 localization to the nucleus. Combined with other reports showing preferential binding of E1A to repressive E2F/DP complexes, this could provide another potential molecular mechanism.

Minor:

1. ADWT is not described.
2. Line 39. Deleted "even".
3. Line 47. Since should be "for".
4. The first viral gene expressed is VA RNAs not E1A. E1A is the first protein coding gene (see

Crisostomo et al, 2019).

5. Line 115. MemberS.

6. Line 172 (and others). Space missing between "E2F1" and "is".

7. Line 173. What mechanism?

8. Line 202. dl348 should not be called wild-type since it is not.

Reviewer #2 (Remarks to the Author):

In this study, the authors reported that a combination of CDK4/6 inhibitors significantly enhanced the replication of oncolytic adenoviruses, and the reduction of RB expression level by CDK4/6 inhibitors played an important role of the synergistic cytotoxicity. The many experiments were well done with various constructs, and the findings revealed the mechanism of the combination therapy to a certain extent. However, several concerns should be addressed before consideration for publication. Especially, the further clarification about the mechanism of CDK4/6 inhibitors-mediated viral replication via RB/E2F1, and the relationship of RB-E2F1 pathway with the crosstalk between E1 and E2 genes may be helpful to understand the underlying mechanism of combination therapy.

1. RB suppresses the virus replication. The effect of CDK4/6 inhibitors was due to the downregulation of RB, whereas the existence of RB protein was necessary for the CDK4/6 inhibitor-mediated enhancement of viral replication (Fig. 1c and Fig. 5c). Why did the suppression of RB by siRNA mitigate the CDK4/6 inhibitor-mediated enhancement of viral replication (Fig. 5c)? Is there a threshold of RB expression level to induce CDK4/6 inhibitor-mediated viral replication?

2. The CDK4/6 inhibitor significantly enhanced the viral DNA replication of ADWT compared to XVir-N-31 (Fig. 2c), whereas the enhancement of particle formation by CDK4/6 inhibitors was almost same for ADWT and XVir-N-31 (Fig. 2d). Why does this discrepancy occur?

3. The Hek293 cells express the E1A gene, which suppresses the RB gene. However, the viral replication was not recognized even in control Hek293 cells, and the CDK4/6 inhibitor did not enhance the viral replication (Fig. 3d). Please explain the reason.

4. The authors stated that E2F1 is important for E2 early promoter activity (Fig. 6c) (lines 203-204). On the other hand, the combination of CDK4/6 inhibitor with adenovirus significantly downregulated the E2F1 protein level (Fig. 4c) at 12 and 24 hpi, whereas the CDK4/6 inhibitor enhanced expression level of E1A and E2A of adenovirus at 12 and 24 hpi (Fig. 2f). It would be valuable to explain this point.

5. Although the E2F1 is important for activating E2 early promoter (Fig. 6c), the E2F1 may not to be so important in adenoviral replication and particle formation (Figs. 5a and b). Please explain the reason. To understand the role of E2F1 in the CDK4/6 inhibitor-mediated enhancement of virus replication, it is necessary to examine the effect of siE2F1 on viral replication in the presence of CDK4/6 inhibitors.

6. The E1A gene of adenovirus significantly enhanced the activity of E2 early promoter (Fig. 6c). Is this E1A-mediated enhancement related with RB/E2F1 pathway?

7. The CDK4/6 inhibitor seems to downregulate the E2 early promoter activity in dl348 ADWT-infected cells (Fig. 6c), whereas the inhibitor enhanced the expression of E2 early transcripts of ADWT (Fig. 4e). Is this discrepancy due to the difference of examined time points (24 and 48 hpi)?

8. The E2 early promoter activity induced by ADWT was higher in pE2-early-luc-TrapM-transfected cells than pE2-early-luc-Trap-transfected cells (Fig. 6e). However, the viral replication is higher in pE2-early-luc-Trap cells (Fig. 6f), and actually ADWT/TR rapidly replicate than ADWT (Figs. 6g and h). Although the authors mentioned this phenomenon in Discussion section, it would be valuable to explain more the reason about this unexpected results.

9. The authors showed that CDK4/6 inhibitor enhanced the viral replication in the tumor xenograft of mice (Fig. 8e) and a strong in vivo antitumor effect. To show the viral replication leads to the antitumor effect, it would be valuable to examine the time course of viral replication in tumors. Furthermore, it would be important to show the data of higher percentage of tumor cell degeneration and necrosis after combination therapy (Fig. 8f).

10. The CDK4/6 inhibitors enhanced the viral replication in the cells with RB gene, indicating that the adenovirus may more rapidly replicate in normal cells. To address the safety issue of the combination therapy, it would be useful to examine the toxicity in normal cells and organ damage in the mice treated with the combination therapy.

Reviewer #3 (Remarks to the Author):

The study would have been much stronger and more relevant with an orthotopic patient sarcoma mouse model (Oncotarget 7, 47556-47564, 2016; Cancer Letters 469, 332-339, 2020).

We would like to thank the reviewer for their thorough reading of our manuscript and addressed the raised concerns. We think that we could significantly improve the manuscript and implemented novel results in the revised version.

We answer in the following letter criticism and suggestions point by point.

Reviewer 1:

Comment 1. Line 64. What about contribution of free E2F1 to apoptosis that would be expected under inactive/degraded Rb condition? This should be discussed in the intro, and is likely contributing to some of the observed effects especially with wt virus, but likely the mutant as well.

Response:

There are two aspects in this question we might have to clarify first. It is in the context of this study important to note that we do not study free E2F1 but study the combination of CDK4/6 inhibitors and an oncolytic adenovirus. That means that the cellular response is not based on one molecule but by the manipulation of a complete signaling cascade.

The second aspect is that the dominant effect of using CDK4/6 inhibitors as monotherapy is the induction of cell cycle arrest in G0/G1. The vast majority of publications report an induction of cell cycle arrest or senescence and only in a very few publications have reported induction of apoptosis using concentrations not achievable in vivo. However, even senescence is only induced in specific cell lines and tumor entities in particular breast cancer. As for a summary, we refer to the following reviews [1-3].

We included in the introduction:

The dominant effect of CDK4/6is as monotherapy is the induction of cell cycle arrest in G0/G1 or senescence but not apoptosis [1-3].

In order to address the reviewers point also in our cellular system, we additionally performed Caspase 3/7 and also tryphan blue staining in a concentration dependent kinetic in the cell lines used in our study. No increase in apoptotic cells could be observed. It is quite the opposite, that the level of activated Caspase 3/7 is downregulated in treated cells, which is understandable since cells are arrested in G0/G1 phase and has been also shown by other groups [4].

We included the new figure in Results and supplements and changed the text in:
CDK4/6is do not induce apoptosis in these cell lines (Supplement 1d,e).

We also include in the Methods section:

Caspase 3/7 and cell death assays:

To determine caspase-dependent apoptosis after treatment with Palbociclib, 2000 cells per well were seeded and grown overnight in 96-well plates. Triplicates were treated with increasing concentrations of the inhibitors for indicated amount of time. Caspase-Glo 3/7 (Promega G8091) and Cell Titer-Blue (Promega G8081) assays were conducted in parallel according to the manufacturer's instructions. The caspase 3/7 activity from the Caspase-Glo assay was normalised to the number of cells present in each condition as determined by Cell Titer-Blue assay. In parallel, 2×10^6 were seeded in 10-cm plates overnight and treated with increasing concentrations of the inhibitors. Medium was aspirated and cells were harvested and stained with Trypan Blue. Stained cells were visualized and quantified using Invitrogen EVOS M5000 microscope and counting 10x optical fields.

In addition, we detected in around 20% of cells in the cell lines used in this manuscript the induction of senescence. We are not showing these results because they are part of another manuscript and not important for this manuscript.

The observation that viral particle formation increases upon treatment has been linked in figure 3 to the very interesting observation that in the combination therapy more cells are producing virus particles. This is a very important aspect in this therapy, since this allows lower viral titer to kill more cells.

Comment 2. Fig 1c. Is there an infection here? If showing selectivity here is the objective, then the authors need to show that ADWT(dl309) can replicate. Furthermore, the authors should show that these cells are getting infected properly and that they're expressing at least E1A if not the other viral genes.

In all experiments shown in this manuscript, we used a MOI that would result in 10-20% cell killing. We explain this point in the criticism under point 5 in detail. In brief this was done to show the difference between the combination therapy and virus monotherapy. We included additionally in supplement figure 1a-b the kinetic between MOI and quantified cell killing and **included the following sentence in results:** We explored this approach in bladder cancer cells and first examined infectivity and the relationship of MOIs used and cell lysis

(supplement 1a-b) [5]. For the examination of combination therapies, we used a MOI that resulted in cell lysis rates of 10-30% throughout this manuscript.

Comment 3. Fig 1e and numerous others. Assaying for viral genomes is not the same as for viral particles, therefore this is not accurate. For example, most rodent cells will replicate human adenovirus DNA but no infectious particles will be made so quantifying vDNA replication and equating it to viral replication is wholly incorrect. The authors should show actual viral replication from viruses not genomes extracted from infected cells. Whether this is done thru PCR or plaque assay doesn't matter, but equating viral genomes in infected cells to viral replication is wrong.

We changed in the figure descriptions and in the text the expression to “viral genome replication”.

Comment 4. Fig 1d. Why is the level of Chk1 vastly different at no drug (0 concentration) between UCN-01 and AZD7762 treated cells? At time 0 there should be no treatment and no difference.

We thank the reviewer for this question. We repeated the experiment again in 3 biological replicates and confirmed that the CHK1 level are not different in the control setting. Thus we changed the WB accordingly in Fig. 1d.

Comment 5. Fig 2a. Why is there no replication of ADWT with no drug treatment? This virus is expected to replicate regardless, but here it is shown not to or very little. This makes no sense whatsoever. Therefore, a control showing that these cells are infected and producing viral proteins is needed. Also, the authors need to show how long it takes the cells to be killed by the infection as compared to treated cells, which seem to die by day 4? Regardless, the WT virus should always kill the cells, so I do not understand why it is not.

This observation is the really exciting one in our manuscript and we are sorry that we explained the result obviously not clear enough. We originally infected cells with a MOI that would result in 50-80% cell killing. However, when comparing monotherapy to combination therapy, an additional effect in the combination therapy was then only about another 20-30%. Thus, we decided to lower the MOI to a level, where one could just measure an effect of 10-

30% cell killing and applied this MOI to all experiments where we tested a combination therapy.

Thus, and we would like to stress this point, the ADWT (and of course also XVir-N-31) infection without the inhibitor results in only about 10-30% in cell lysis in the MOIs used whereas in the combination with a CDK4/6 inhibitor we observe this dramatic effect of 90% cell lysis.

As also pointed out in the discussion, this observation is for a therapy of great interest because it means that even in a suboptimal infection, a virus would replicate very efficiently and would still result in tumor cell lysis. It would also indicate that one could decrease the general MOI used for patient treatment. This aspect of the therapy was also addressed in figure 3 in which we show that in the combination more cells produce virus. We also included now in supplement figure 1a-b a kinetic showing cell killing and MOI.

Comment 6. Fig 2g. ADWT should be shown in these cells as well to indicate that there is some degree of selectivity. Furthermore, if the XVir-N-31 is selective oncolytic virus, it needs to be shown that it does not kill normal cells with cdk4/6 inhibition, which is not shown anywhere in the manuscript.

In this figure, we show again that one requirement for the combination therapy is the endogenous expression of an active Rb protein. The cancer selectivity of XVir-N-31 is not subject of the manuscript and has been examined not only in vitro/in vivo [6] but also within a clinical trial toxicity study. This toxicity study in the context of a clinical trial in glioblastoma patients, funded by the German BMBF, was performed with more than 500 hundred Syrian Hamsters showed no signs of toxicity or replication (unpublished data). Since YB-1 localization is not affected by CDK 4/6 treatment XVir-N-31 (figure 3D) and CDK 4/6 treatment of normal cells reduces viral replication [7] we think that it would be a redundant result so that we have not to address this issue in detail in this paper again.

Comment 7. Fig 2f. Why is there only 1 E1A band in the ADWT lanes? E1A is known to produce 5 isoforms, 4 of which are easily detectable during infection by western.

There are several bands detected in the WB directed against E1A but they are not ideally resolved in size so that one has to have a very close look to the western blot shown. To study E1A isoforms one would have to use gradient gels as done for example in a publication by Soriano et al., 2019 [8]. In our view for analyzing the combination therapy approach it is not

necessary to study the different isoforms but to show the effects of CDK4/6is in a time kinetic for E1A expression level.

Comment 8. Fig 2f and a. Hexon production also indicated that viruses are being assembled, particularly with ADWT, yet cell viability is little compromised in (a). What is the reason behind this as it doesn't add up? Generally, once hexon production is initiated the obvious CPE is visible, was cellular morphology altered in (a)?

As we already described when addressing points 2 and 5 raised by the reviewer, we used a very low MOI in order to detect the real difference in the lytic capacity of this combination therapy. We used in the experimental design these special conditions in order to better resolve the real potency of this combination therapy.

Of course, the assumption is correct that a change in morphology was observed also in (a).

We included in the results section for Figure 1 that we worked with very low MOI and show now a cell killing kinetic using different MOIs in supplement 1a-b).

Comment 9. Line 148. What is the level of CAR expression on the different cell lines? The authors need to show this either by western, or qRT-PCR at the least.

This is an important aspect raised and our group published these data already in 2018 (Lichtenegger et al 2018, figure 1B and figure 2) but did not include this reference at this point in the manuscript.

We now included this information in Results:

We explored this approach in bladder cancer cells and first examined infectivity and the relationship of MOIs used and cell lysis (supplement 1a-b) [5]. For the examination of combination therapies, we used a MOI that resulted in the control in toxicities of 10-30% throughout this manuscript.

Comment 10. Fig 3d. The observation of lack of enhancement of infection/replication in 293 cells versus the others suggests that this may entirely be an E2F1-apoptosis driven mechanism. Inhibition of cdks would preclude E2F release from pRb, and prevent activation of pro-apoptotic genes by free E2F1, in 293 cells, that constitutively express E1A, this would have little effect as Rb is already inactivated by E1A and further inactivation would have no to little effect. This possibility was not investigated by the authors whatsoever and it needs to be, it is a critical aspect of the potential mechanism.

293 cells do not respond to CDK4/6 inhibitors and we show in the new supplement 2a that there is no decrease of RB or phosphorylated RB upon treatment with PD. We totally agree that one would in such a situation not expect any change on the replication of the virus when combining it with a CDK4/6 inhibitor. This result strongly supports our hypothesis that the molecular mechanism of this combination therapy is dependent on the modification of RB level and thus responsiveness to CDK4/6 inhibitors. Again, apoptosis does not play a role because an apoptotic cell does not provide a good environment for viral replication. Also, in our cell culture, trypan blue staining revealed 100% healthy 293 cells (Figure 1e). As shown by the group of F.A. Dick in 2008, “E2F1 functions to maintain cell viability in the response to E1A expression” which means that E2F1, induced/stabilized by an adenovirus, is essential for maintaining a functional and living cell that is able to generate novel virus.

Further, the reviewer is asking for the “critical aspect of the potential mechanism by RB-E2F1”. We examined the putative mechanism in detail in figures 4-7.

Comment 11. Fig 4c. What are the levels of the other E2Fs and other pocket proteins (p107 and p130), particularly p130 which can often substitute of pRb under certain pRb-deficient conditions.

We included now western blots against p107 and p130 in figure 4c. The expression level of E2Fs were already shown in figure 2F.

We changed the text in:

We next examined protein expression of RB, p107, p130 and E2F1 upon treatment with PD and XVir-N-31 or ADWT as mono- and combination therapy in a time kinetic. Downregulation of RB, p107 and E2F1 protein was observed with 500nM PD monotherapy that remained unchanged over 36 hours. No change in expression level of p130 could be detected.

We included in Figure 5 d the novel data and changed the text in:

We also transfected SK-N-MC cells that are resistant to treatment with CDK4/6i due to a point mutation in RB⁴² with either siRNAs directed against E2F1, RB, p107 or p130. Only with an siRNA against RB we observed a significant 4 fold increase in viral genome replication (Fig. 5d).

Comment 12. Fig 4d. Why is there a drop of E1A/E2 levels? This should go up continuously, especially E2 as was previously shown by Crisostomo et al 2019. If this is

due to the way the data is presented, I think a better and more informative way would be to present this relative to a housekeeping gene rather than fold change versus wt.

We thank the reviewer for this question because we discussed the way to present the dataset in this figure before. What is shown is the relative expression level to control because we wanted to emphasize the earlier expression of E1A. However, if one shows relative gene expression normalized to GAPDH, the curve is constantly increasing. Thus we decided now to change the figure back to gene expression relative to GAPDH.

Comment 13. Fig 5. Blots of knockdown and/or overexpression must be shown for all genes.

We included these data for all experiments in which we manipulated protein level in supplementary figure 3.

Comment 14. Fig 5c. Is the reduced genome replication due to inability of E1A to be recruited to pRb-repressed promoters in the presumed absence of pRb?

This question is difficult to understand since E1A is not directly recruited to promoters but affects cellular proteins. The T24shRB cells do respond substantially lesser to PD treatment relative to control T24shctrl cells. Once we treat the cells with PD, the effect on replication is relative to the control (T24shctrl) also greatly attenuated in the T24shRB cells, thus supporting our hypothesis that modulation of the RB expression level is an essential molecular parameter in this combination therapy.

Comment 15. Line 212-213, supplementary figure. This is insufficient as changing sequence is not likely to affect mRNA expression but might affect protein sequence/stability, etc. Therefore, the authors need to show that viral proteins (L4-encoded) are not compromised in expression by the introduction of these mutations.

This is an important question. In this figure we analyzed regulation of gene expression dependent of the E2F1 binding site in the E2 early promoter. Surprisingly, this mutation affects E1A expression which in turn influences the expression of late genes. We think that a detailed study of this mutation goes beyond this finding and should not part of this manuscript. However, it is highly unlikely that a structural virus protein expressed on a later stage in the viral life cycle has an effects on viral replication at early time points.

Comment 16. Line 214. Genome replication, not virus. Also, was the level of E2 proteins affected by this mutant as compared to wt? This needs to be shown.

We do not understand this question because we examined protein level of E1A, E2A and Hexon already in Figure 7b, comparing ADWT with ADWT/E2Fm.

Comment 17. Line 219-220. Assuming that E2F1 is the only E2F that will bind to these sites is grossly simplistic, furthermore, it totally ignores the important role of E1A in transactivation of ALL viral promoters.

We think that there is a misunderstanding of our text. We never claim that the novel developed “trap” sequence only binds to E2F1. When reading the manuscript, please recognize that we are talking about the possibility to also study the putative role of E2F1 but that the sequence we cloned contains putative E2F binding sites!

As to the second part of the comment: With this construct, the attempt was to sequester free E2Fs as described in the cited reference. That E1A is the central player in the viral life cycle was not in doubt and has not been studied with this construct.

Comment 18. Fig 7c. This experiment is missing the important IgG negative control and the data should be represented as % of input rather than copy number.

We thank the reviewer for this valuable suggestion. The effect between the different constructs used is now much better to recognize. We included also the IgG control in Figure 7c and changed the text accordingly.

Comment 19. Fig 8. Where is the ADWT in these studies?

Our research is focusing on the development of novel treatment options for cancer using a described and by German official authorities accepted oncolytic adenovirus (XVir-N-31). This manuscript is about application of this oncolytic virus in combination with a small molecule inhibitor against CDK4/6. Since ADWT is not a conditionally replicating oncolytic adenovirus, we only tested XVir-N-31 in a murine model and it is surprising, that the combination therapy has such a substantial impact.

ADWT would never be used in a clinical trial so that an animal experiment would only result in an unnecessary consumption of mice but would not provide further insights into therapeutic options.

Comment 20. Discussion. What happens when the Rb-binding site of E1A is deleted?

This is an important question, which has been addressed by Zheng et al 2016 as cited in our manuscript. This group demonstrated that this binding site causes repression of E1A in normal cells and thereby consequently virus genome replication. They stated “PD0332991 (Pabociclib), a specific CDK4/6 inhibitor, dephosphorylates pocket proteins to promote their interaction with E2Fs and inhibited wild-type Ad5 replication dependent on the conserved E2F binding site”. Importantly, that means that the addition of palbociclib will not raise any safety concern regarding XVir-N-31 when used in this combination therapy.

Comment 21. Discussion. I suspect that inhibition of cdk4/6 results in enhanced complex of E2F/pRb, which allows greater E1A access to certain promoters via pRb, or perhaps directly through E2F. This should be determined. The authors need to show whether inhibition of cdk4/6 stabilizes the E2F/pRb complex and whether E1A is present in this complex during infection, and whether it accesses promoters through this interaction. The authors also fail to look at other E2Fs, particularly repressive ones that E1A is known to bind to preferentially and directly (via heterodimerized DP-1).

Since we and others demonstrate that treatment of cells and tumours result in degradation of pRb and E2Fs, this complex is only detectable in very low concentrations upon treatment (as for review see [2]). We show in figure 4c this downregulation again and published those findings also in previous own publications in cell lines used in this manuscript ([9], [10]) confirming results from dozens of other groups. We do not question the binding of E1A to the pRb/E2F complex but describe that the combination therapy results in a downregulation of this complex and by that facilitate replication and cellular lysis.

As to the second part of this comment: in a publication from F.A. Dick in 2008 published in the Journal of Virology, they analyzed binding of E1A to RB and E2Fs and show that “endogenous pRB readily interacts with E2F1 in the presence of E1A, but not with other E2Fs”.

We performed in addition experiments with siRNAs directed against E2F3 and -4 since we observed upregulation of these E2Fs upon CDK4/6i treatment (see figure 2F). However, viral genome replication was not substantially affected by downregulation of any other E2F.

siRNA experiments with siRNAs against E2F1,3 and -4 have been included in fig 5A (genome replication) and B (particle formation). The text has changed into:

In order to clarify whether the E2Fs affected by CDK4/6is or RB proteins are the crucial factors for the improved viral genome replication, we first down regulated E2F1,-3,-4 and

RB1, RBL1 (p107), RBL2 (p130) proteins by siRNA technology (supplement 2a-e). Silencing of the E2F family members 1,3 and 4 that were upregulated by CDK4/6i treatment (Figure 2e) in T24 cells resulted in a slight increase in viral genome replication and as for E2F4 also viral particle formation (Figure 5a, 5b) without reaching significance, indicating that these E2Fs are probably involved but not the decisive factor for the synergistic effect of the combination therapy.

Comment 22. Line 319. This is consistent with E1A having more access to E2Fs in the absence of pRb.

Please see answer to comment 21.

Comment 23. Line 334. Authors claim that only pRb is important is not based on evidence, as the manuscript never investigated the role of p107 nor p130.

We included these data now in figure 4c and 5d and changed the text accordingly.

Comment 24. Discussion. There is no discussion or investigation of apoptosis driven by high E2F1 activity. This needs investigation.

As already pointed out, an increase in apoptosis would cause a drop in viral particle formation what has been published by many groups and is thus common knowledge by experts in the field.

As shown in the manuscript (figure 2f), E2F1 expression in combination with CDK 4/6 inhibition is decreased. As we now additionally included (see supplement figure 1d,e), apoptotic marker are even decreased upon treatment.

Comment 25. Statistics are largely missing from most of the data figures.

We have included statistics where applicable.

Comment 26. The role of E4orf6/7 in nuclear translocation of E2F4 is not mentioned or investigated. Since E4 is driven by E1A, E1A lacking CR3 will drive this gene poorly, leading to lower levels of E4 proteins (at least early in infection), which could affect E2F4 localization to the nucleus. Combined with other reports showing preferential binding of E1A to repressive E2F/DP complexes, this could provide another potential molecular mechanism.

The reviewer is of course correct with the assumption that viruses lacking CR3 (such as XVir-N-31) will replicate relative to ADWT slightly less effectively due to the lower transactivating capacity on adenoviral genes, actually published by our group before [11].

However, this aspect has already been investigated in one important report [12] regarding E2F4 in which the authors state: “In the present study, we show that E1A 13S directly binds to E2F/DP-1 complexes, likely via association with DP-1. This binding results in activation of E2F-regulated genes, with E2F4 and E2F5 being activated the most strongly. Activation is restricted largely to E1A 13S, while E1A 12S has a minimal effect. We also show that E1A is directly recruited to E2F-regulated promoters during viral infection and that E1A 13S, but not E1A 12S, can enhance E2F4 promoter occupancy”. These results add substantially to our understanding of how E1A can deregulate the cell cycle and highlight important differences between E1A 12S and E1A 13S function. In addition, deletion of E4orf6/7 has only little impact on viral replication [13]. Thus, in our view these results could not provide another potential molecular mechanism.

Thus, since XVir-N-31 (expressing only E1A12S) strongly responds to CDK 4/6 inhibition, we concluded that E2F4 could not provide us with another potential molecular mechanism.

Minor:

Comment 1. ADWT is not described

We included in Material and Methods: **We used dl309 (designated as ADWT in the manuscript) as control (Bett et al., 1995, Virus research 39, 75-82).**

Due to deletions of adenovirus DNA between 30005-30705 no E3 14.7K, 14.5K and 10.4K protein are expressed but E1A expression is not affected.

Comment 2. Line 39. Deleted “even”.

Has been done.

Comment 3. Line 47. Since should be “for”.

Has been done

Comment 4. The first viral gene expressed is VA RNAs not E1A. E1A is the first protein coding gene (see Crisostomo et al, 2019).

We write now: **Adenoviral replication has been examined for decades [14] and is tightly linked to an interplay with the viral master regulator protein E1A, one of the first viral genes expressed after infection [15],**

Comment 5. Line 115. MemberS.

Comment 6. Line 172 (and others). Space missing between “E2F1” and “is”.

Comment 7. Line 173. What mechanism?

Comment 8. Line 202. dl348 should not be called wild-type since it is not.

We have corrected these minor points in the revised version.

Reviewer #2 (Remarks to the Author):

In this study, the authors reported that a combination of CDK4/6 inhibitors significantly enhanced the replication of oncolytic adenoviruses, and the reduction of RB expression level by CDK4/6 inhibitors played an important role of the synergistic cytotoxicity. The many experiments were well done with various constructs, and the findings revealed the mechanism of the combination therapy to a certain extent. However, several concerns should be addressed before consideration for publication. Especially, the further clarification about the mechanism of CDK4/6 inhibitors-mediated viral replication via RB/E2F1, and the relationship of RB-E2F1 pathway with the crosstalk between E1 and E2 genes may be helpful to understand the underlying mechanism of combination therapy.

Comment 1. RB suppresses the virus replication. The effect of CDK4/6 inhibitors was due to the downregulation of RB, whereas the existence of RB protein was necessary for the CDK4/6 inhibitor-mediated enhancement of viral replication (Fig. 1c and Fig. 5c). Why did the suppression of RB by siRNA mitigate the CDK4/6 inhibitor-mediated enhancement of viral replication (Fig. 5c)? Is there a threshold of RB expression level to induce CDK4/6 inhibitor-mediated viral replication?

This is a very interesting question since it would argue either for a tipping point or a linear process based on availability of RB. We performed a concentration kinetic and used as a readout either a western blot for detecting RB level or a potency assay to detect the viral effect on cell lysis (see additional figure in supplement). The result shows a linear effect, meaning the virus induced cell lysis directly correlates with the reduction in RB level.

We included these data in Figure 2 and write in the results part: We first tested in a dose dependent kinetic the correlation of PD treatment, RB expression level and cell lysis and could show, that with the molecular response to PD an increase in cell killing was induced by

XVir-N-31 (Fig. b,c). The improvement of virus induced cell lysis therefore correlates with expression level of RB but it does not require a complete elimination of this protein.

In Figure 5e (former 5c), we used a lentiviral transduced cell line with a shRNA against RB. These T24shRB cells do respond substantially lesser to PD treatment relative to control T24shctrl cells. Once we treat the cells with PD, the effect on replication is relative to the control (T24shctrl) also greatly mitigated in the T24shRB cells, thus supporting our hypothesis that modulation of the RB expression level is an essential molecular parameter in this combination therapy. However, please note that in shRB expressing cells viral genome replication is still enhanced compare to the control cell counterpart.

Comment 2. The CDK4/6 inhibitor significantly enhanced the viral DNA replication of ADWT compared to XVir-N-31 (Fig. 2c), whereas the enhancement of particle formation by CDK4/6 inhibitors was almost same for ADWT and XVir-N-31 (Fig. 2d). Why does this discrepancy occur?

We showed in figure 2c (now it is 2d) only XVir-N-31 and examined in a time kinetic the genomic replication comparing control treated cells with PD treated cells. The direct matching data with ADWT are shown in supplement Figure 1a and h.

Comment 3. The Hek293 cells express the E1A gene, which suppresses the RB gene. However, the viral replication was not recognized even in control Hek293 cells, and the CDK4/6 inhibitor did not enhance the viral replication (Fig. 3d). Please explain the reason.

We included a novel western blot that shows expression level of RB and phosphorylated RB in ctrl and PD treated HEK 293 cells in supplement 2a. RB protein is abundantly expressed in these cells but CDK 4/6 inhibitors do not affect RB expression and phosphorylation level. The cell cycle state is also not affected by CDK 4/6 inhibitors in this cell line. Thus we show that CDK4/6 inhibitors increase adenoviral genome replication in cells that do respond to these inhibitors by a reduction in RB protein level and also on a functional level.

We believe that E1A present in Hek293 cells protects somehow the proteolytic degradation of RB and rather stabilize RB in these cells as described in these to manuscripts [16, 17].

We included data in supplementary 2a and changed the text in:

This result was extended to three different bladder cancer derived cell lines (UM-UC-3, T24, RT112) and Hek293 cells as control since they do not respond to CDK4/6is although they express RB protein (supplement 2a).

Comment 4. The authors stated that E2F1 is important for E2 early promoter activity (Fig. 6c) (lines 203-204). On the other hand, the combination of CDK4/6 inhibitor with adenovirus significantly downregulated the E2F1 protein level (Fig. 4c) at 12 and 24 hpi, whereas the CDK4/6 inhibitor enhanced expression level of E1A and E2A of adenovirus at 12 and 24 hpi (Fig. 2f). It would be valuable to explain this point.

We thank the reviewer for addressing this point which is critical. When looking only on E2F1 level one could imagine that E2F1 might act as a suppressor of viral replication at early time points. However, this interpretation would completely ignore the current model and the role of E2F1 in viral life cycle which has been developed based on many excellent studies. Also, E2F activity is always regulated dependent on its complexation with other proteins such as p300 and others.

This challenge in our data prompted us to perform the western blots in figure 4c and we observe that in the presence of virus, E2F1 downregulation recovers already at 12 hours after infection whereas RB level are kept down and this only in the presence of virus.

We address this point now by writing: One explanation might be that RB suppression by CDK4/6is does release the activity of remaining E2F1 that initiates in turn expression of E1A. The data presented in the CHIP assay (Fig 7c) indicate that the E2F binding sites in E2-Early and the E1A enhancer compete for available E2F. Thus, our hypothesis is that CDK4/6 inhibitors although decreasing the level of both, RB and E2F1 result in a pool of “free” E2F that is first directed to the E1A-enhancer and initiate expression of this essential protein for replication.

Comment 5. Although the E2F1 is important for activating E2 early promoter (Fig. 6c), the E2F1 may not be so important in adenoviral replication and particle formation (Figs. 5a and b). Please explain the reason. To understand the role of E2F1 in the CDK4/6 inhibitor-mediated enhancement of virus replication, it is necessary to examine the effect of siE2F1 on viral replication in the presence of CDK4/6 inhibitors.

We included a novel dataset in Figure 5c and replaced the text in:

In order to clarify whether the E2Fs affected by CDK4/6is or RB proteins are the crucial factors for the improved viral genome replication, we first down regulated E2F1,-3,-4 and

RB1, RBL1 (p107), RBL2 (p130) proteins by siRNA technology (supplement 3a-e). Silencing of the E2F family members 1,3 and 4 that were upregulated by CDK4/6i treatment (Figure 2f) in T24 cells resulted in a slight increase in viral genome replication and as for E2F4 also viral particle formation without reaching statistical significance (Figure 5a, 5b), indicating that these E2Fs are probably involved but not the only decisive factor for the synergistic effect of the combination therapy. Also in the combination with PD, E2F1 silencing did not contribute to a better response to viral genome replication or particle formation (Fig 5c).

We would like to point out that the E2-early promoter might contain additional binding motives besides E2F-binding sites, which also influence viral replication. In addition, particle formation is a tightly and highly regulated process where many other viral proteins are involved. Likewise, deletion of E1B55k or E4orf6 have a significant impact of viral particle formation.

Comment 6. The E1A gene of adenovirus significantly enhanced the activity of E2 early promoter (Fig. 6c). Is this E1A-mediated enhancement related with RB/E2F1 pathway?

The presence of E1A releases E2F from its inactive status in the complex with RB as described also by [18].

Comment 7. The CDK4/6 inhibitor seems to downregulate the E2 early promoter activity in dl348 ADWT-infected cells (Fig. 6c), whereas the inhibitor enhanced the expression of E2 early transcripts of ADWT (Fig. 4e). Is this discrepancy due to the difference of examined time points (24 and 48 hpi)?

The difference in these two figures are inherent to the system used. In figure 6c we use a promoter construct fused to the reporter gene in a plasmid. This is a simple design and should result in a signal, induced by cellular E2F proteins. Thus, we would expect that CDK 4/6 inhibitors due to their function inhibit cellular E2F regulated genes in a similar way as genes expressed by a plasmid.

This appears to be other way around for E2F regulated genes present in the viral genome. This is the novelty of the manuscript and this is based on the inhibition of Rb expression. Thus, the discrepancy in time points are not the reason for the observed effects.

We wrote additionally in the results section: These results demonstrate that these novel constructs respond to cellular factors and the presence of E1A. However, it is important to

note that these assays also demonstrate that transcription of E2-early genes in the context of the viral genome are controlled by additional factors since we show in Figure 4e that PD treatment improves transcription of E2-early.

Comment 8. The E2 early promoter activity induced by ADWT was higher in pE2-early-luc-TrapM-transfected cells than pE2-early-luc-Trap-transfected cells (Fig. 6e). However, the viral replication is higher in pE2-early-luc-Trap cells (Fig. 6f), and actually ADWT/TR rapidly replicate than ADWT (Figs. 6g and h). Although the authors mentioned this phenomenon in Discussion section, it would be valuable to explain more the reason about this unexpected result.

This is a complicated experiment and we obviously did not describe it sufficiently. In these figures, we use a plasmid with a normal wt E2-early luciferase reporter cassette and cloned additionally the mentioned E2F-Trap site or the E2F-TrapM site as a second cassette in the vector. We had two ideas in mind when doing this: 1st The idea was that we could “trap” the E2Fs and thus downregulate expression of E2-early. 2nd We wanted to test, if trapping E2Fs have an effect on replication. In figure 6e, we received the expected result on regulation of the E2-early reporter construct. By trapping the complex RB/E2F, ADWT infection results in a lower activation of the E2-early promoter compared to the mutated Trap construct. Measuring viral replication using the same plasmid constructs show an increase in viral replication in pTrap transfected cells compared to pTrapM transfected cells. We then tested this result in Figure 6g, by developing a novel adenoviral vector, in which we included this E2F trapping sequence in the E3 region and observed the same result.

We changed the text in the manuscript and hope that it now explains the construct and results better:

The combined infection and transfection of cells with ADWT and either pE2-early-luc-Trap or pE2-early-luc-TrapM showed that the additional E2-trap cassette is sufficient to suppress E2-early luciferase expression. As expected, PD treatment had no influence in this construct. However, with the mutated Trap-construct E2F protein complexes are not sequestered so that we observed a significant increase in Luciferase expression. PD treatment also slightly reduced activity of the E2-early luciferase cassette (Fig. 6e).

In regard to the second question in this comment:

Our results might again question if E2F-protein complexes could have also negative effects on viral replication. However, we do not want to stress this thought in the manuscript too much,

because that would need very thorough additional experiments to compare previous excellent results from other groups and our observations.

Comment 9. The authors showed that CDK4/6 inhibitor enhanced the viral replication in the tumor xenograft of mice (Fig. 8e) and a strong in vivo antitumor effect. To show the viral replication leads to the antitumor effect, it would be valuable to examine the time course of viral replication in tumors. Furthermore, it would be important to show the data of higher percentage of tumor cell degeneration and necrosis after combination therapy (Fig. 8f).

A more detailed study of this combination approach is currently performed. However, this will also include the participation of the immune system since CDK 4/6 inhibitors processes also immune-activating properties, which play a central role in reaching a clinical meaningful efficacy. Since this is a very large project including different complicated mouse models, we believe that the presented in vivo results support our novel finding, that decreasing RB expression facilitate viral genome replication and tumor cell lysis.

Comment 10. The CDK4/6 inhibitors enhanced the viral replication in the cells with RB gene, indicating that the adenovirus may more rapidly replicate in normal cells. To address the safety issue of the combination therapy, it would be useful to examine the toxicity in normal cells and organ damage in the mice treated with the combination therapy.

This statement is absolutely correct for cancer cells. For normal cells, there are only limited data available that study effects on cell cycle and RB expression. However, general toxicology studies were conducted with oral palbociclib administration for up to 27 weeks in rats and 39 weeks in dogs

https://www.accessdata.fda.gov/drugsatfda_docs/nda/2015/207103Orig1s000PharmR.pdf. An application of oncolytic viruses over a so long period is not planned in our phase I study. In addition only local application of the virus is planned minimizing a systemic distribution of the virus. In addition, CDK 4/6 inhibitors have been recently described as novel protectors against cancer therapy induced toxicity [19].

We received funding from the German BMBF for a phase I clinical trial in which we will implement after consultation with the German authorities (Paul-Ehrlich Institute, (PEI) a toxicity study in Syrian Hamster, which is an accepted model to address this issue.

Reviewer #3 (Remarks to the Author):

The study would have been much stronger and more relevant with an orthotopic patient sarcoma mouse model (Oncotarget 7, 47556-47564, 2016; Cancer Letters 469, 332-339, 2020).

Dear reviewer, we agree that an orthotopic model is always closer to the real situation. However, we wanted to examine in this approach only if the novel combination therapy compared to monotherapy is improving the effect on a tumor in vivo. We think that for this specific purpose the model used is sufficient.

1. Goel, S., et al., *CDK4/6 Inhibition in Cancer: Beyond Cell Cycle Arrest*. Trends Cell Biol, 2018. **28**(11): p. 911-925.
2. Klein, M.E., et al., *CDK4/6 Inhibitors: The Mechanism of Action May Not Be as Simple as Once Thought*. Cancer Cell, 2018. **34**(1): p. 9-20.
3. Knudsen, E.S. and A.K. Witkiewicz, *The Strange Case of CDK4/6 Inhibitors: Mechanisms, Resistance, and Combination Strategies*. Trends Cancer, 2017. **3**(1): p. 39-55.
4. Wang, Z., et al., *CDK4/6 inhibitor protects against myocardial cells apoptosis by inhibiting RB phosphorylation in H9c2 cells*. Biochem Biophys Res Commun, 2019. **509**(4): p. 949-953.
5. Lichtenegger, E., et al., *The Oncolytic Adenovirus XVir-N-31 as a Novel Therapy in Muscle-Invasive Bladder Cancer*. Hum Gene Ther, 2019. **30**(1): p. 44-56.
6. Mantwill, K., et al., *YB-1 dependent oncolytic adenovirus efficiently inhibits tumor growth of glioma cancer stem like cells*. J Transl Med, 2013. **11**(1): p. 216.
7. Zheng, Y., T. Stamminger, and P. Hearing, *E2F/Rb Family Proteins Mediate Interferon Induced Repression of Adenovirus Immediate Early Transcription to Promote Persistent Viral Infection*. PLoS Pathog, 2016. **12**(1): p. e1005415.
8. Soriano, A.M., et al., *Adenovirus 5 E1A Interacts with E4orf3 To Regulate Viral Chromatin Organization*. J Virol, 2019. **93**(10).
9. Sathe, A., et al., *CDK4/6 Inhibition Controls Proliferation of Bladder Cancer and Transcription of RBI*. J Urol, 2016. **195**(3): p. 771-9.
10. Pan, Q., et al., *CDK4/6 Inhibitors in Cancer Therapy: A Novel Treatment Strategy for Bladder Cancer*. Bladder Cancer, 2017. **3**(2): p. 79-88.
11. Rognoni, E., et al., *Adenovirus-based virotherapy enabled by cellular YB-1 expression in vitro and in vivo*. Cancer Gene Ther, 2009. **16**(10): p. 753-63.
12. Pelka, P., et al., *Adenovirus E1A directly targets the E2F/DP-1 complex*. J Virol, 2011. **85**(17): p. 8841-51.
13. Swaminathan, S. and B. Thimmapaya, *Transactivation of adenovirus E2-early promoter by E1A and E4 6/7 in the context of viral chromosome*. J Mol Biol, 1996. **258**(5): p. 736-46.
14. Berk, A.J., *Recent lessons in gene expression, cell cycle control, and cell biology from adenovirus*. Oncogene, 2005. **24**(52): p. 7673-85.

15. Gallimore, P.H. and A.S. Turnell, *Adenovirus E1A: remodelling the host cell, a life or death experience*. *Oncogene*, 2001. **20**(54): p. 7824-35.
16. Nemajerova, A., et al., *Rb function is required for E1A-induced S-phase checkpoint activation*. *Cell Death Differ*, 2008. **15**(9): p. 1440-9.
17. Seifried, L.A., et al., *pRB-E2F1 complexes are resistant to adenovirus E1A-mediated disruption*. *J Virol*, 2008. **82**(9): p. 4511-20.
18. Frisch, S.M. and J.S. Mymryk, *Adenovirus-5 E1A: paradox and paradigm*. *Nat Rev Mol Cell Biol*, 2002. **3**(6): p. 441-52.
19. Boopathi, E. and C. Thangavel, *CDK4/6 inhibition protects normal cells against cancer therapy-induced damage*. *Transl Cancer Res*, 2020. **9**(2): p. 405-408.

REVIEWER COMMENTS

Reviewer #1 (Remarks to the Author):

The authors have addressed most of my major issues and clarified others. I am satisfied with this and I recommend publication.

Reviewer #2 (Remarks to the Author):

The authors have performed several additional experiments and improved the manuscript. There seems to be still several concerns should be addressed before consideration for publication.

1. Comment 1 in the original review: The experiments of RB level and relative survival according to the concentration of PD (Figs. 2B and 2C) were valuable. On the other hand, in Fig. 5E, it is not still clear why the suppression of RB by siRNA mitigated the CDK4/6 inhibitor-mediated enhancement of viral replication. As the authors' hypothesis, the modulation of the RB expression level seems to be essential in the combination therapy. Please show the RB levels in virus-infected T24shCtr and T24shRB with or without CDK4/6 inhibitor.

2. Comment 3 in the original review: Since the staining of HEK293 cells, in which adenovirus is replication, is faint compared to other bladder cancer cell lines, more intense brown should be presented to show the virus replication in Fig. 2D.

3. Comment 4 in the original review: It is not still clear why the combination of CDK4/6 inhibitor with adenovirus significantly downregulated the E2F1 protein level in Fig. 4C, whereas the CDK4/6 inhibitor enhanced expression level of E2A of adenovirus in Fig. 2G. As the authors' suggestion, CDK4/6 inhibitors may increase a pool of free E2F which is directed to the E1A enhancer and initiate the virus replication. Therefore, it would be valuable to show RB-E2F and free E2F in virus-infected cells treated with or without CDK4/6 inhibitor.

4. Comment 5 in the original review: The experiments using siE2F indicated that the E2F family were not decisive factor for the combination therapy in Figs. 5A and 5B. On the other hand, the replication activity of ADWT/E2Fm was much decreased compared to ADWT in Fig. 6D, which may indicate the importance of E2F1 in the virus replication. Please explain the reason.

5. Comment 8 in the original review: Although the expression of E2 early gene may not be strongly correlated with the virus replication, it is still unclear why the E2 early promoter activity induced by ADWT and the replication of adenovirus were different between in pE2-early-luc-TrapM-transfected and pE2-early-luc-Trap-transfected cells (Figs. 6E and 6F). Since the readers would like to know the reason, please explain the reason to some extent.

6. In addition, the replication activity of ADWT/E2Fm and ADWT/Trap were opposite compared to ADWT in Figs. 6D and 6H. Is this reasonable?

7. Comment 9 in the original review: The authors did not show time course of viral replication in tumors and that CDK4/6 inhibitor enhanced the viral replication in the tumor xenograft of mice and the data of higher percentage of tumor cell degeneration and necrosis after combination therapy. It would be still valuable to show the results to confirm the enhanced virus replication in the tumor by the combination therapy.

8. Comment 11 in the revised version: In this study, the authors showed that the E2F1 is not a crucial factor and RB is important for the mechanism of the combination therapy. Please show the more detailed mechanism clarified in this study including the role of CDK4/6 inhibitors on RB and E2F1 in Figure 9.

Reviewer #3 (Remarks to the Author):

The authors seem not to understand the importance and relevance of orthotopic models of sarcoma to their study.

The study would have been much stronger and more relevant with an orthotopic patient sarcoma mouse model (Oncotarget 7, 47556-47564, 2016; Cancer Letters 469, 332-339,2020). The authors' response to this point is unsatisfactory.

Reviewer #2 (Remarks to the Author):

The authors have performed several additional experiments and improved the manuscript. There seems to be still several concerns should be addressed before consideration for publication.

We would like to thank the reviewer for the excellent questions raised on our data. These ideas definitely improve the manuscript. We addressed these questions and added several additional novel data to the manuscript.

1. Comment 1 in the original review: The experiments of RB level and relative survival according to the concentration of PD (Figs. 2B and 2C) were valuable. On the other hand, in Fig. 5E, it is not still clear why the suppression of RB by siRNA mitigated the CDK4/6 inhibitor-mediated enhancement of viral replication. As the authors' hypothesis, the modulation of the RB expression level seems to be essential in the combination therapy. Please show the RB levels in virus-infected T24shCtr and T24shRB with or without CDK4/6 inhibitor.

We examined the protein level of RB in the T24shCtr and T24shRB cells 24 hours after infection. In the conditions used, there was no change or only very subtle downregulation in RB level in the control cells upon virus infection. However, treatment with PD significantly decreased RB level. The minor effect on RB in the monotherapy using adenovirus is probably due to the low MOI we use in this experiment in order to show the difference in the combination when comparing between PD treated and non treated cells.

In the T24shRB cells, no substantial change in the intensity of the remaining very faint RB band could be detected. We included this figure now in supplement figure 3g and changed the text in the manuscript accordingly into:

“Upon treatment, only the T24Ctrl cells showed changes in RB level, whereas in the T24shRB cells, the remaining RB was not significantly altered (supplement 3g), indicating the mitigated effect of the CDK4/6 inhibitor-mediated enhancement of viral replication”

2. Comment 3 in the original review: Since the staining of HEK293 cells, in which adenovirus is replication, is faint compared to other bladder cancer cell lines, more intense brown should be presented to show the virus replication in Fig. 2D.

We improved the staining intensity and replaced the photo in Fig. 3d.

3. Comment 4 in the original review: It is not still clear why the combination of CDK4/6 inhibitor with adenovirus significantly downregulated the E2F1 protein level in Fig. 4C, whereas the CDK4/6 inhibitor enhanced expression level of E2A of adenovirus in Fig. 2g. As the authors' suggestion, CDK4/6 inhibitors may increase a pool of free E2F which is directed to the E1A enhancer and initiate the virus replication. Therefore, it would be valuable to show RB-E2F and free E2F in virus-infected cells treated with or without CDK4/6 inhibitor.

We agree that this will be valuable and we have tried to show this aspect by establishing an EMSA analysis using a LUEGO system instead of ³²P-labeled nuclear lysates in a Masters thesis. As bait, we used the E2early promoter region. Unfortunately, we were not able to obtain any supershifts when using antibodies directed against E2F1-5 which might be due to a sensitivity issue of the LUEGO system.

What we do show in our data in Figure 4c is the substantial recovery of E2F1 expression level upon infection of cells with adenovirus whereas Rb downregulation is even further downregulated by the virus infection. Since RB is almost completely gone, this observation implies availability of free E2F. Because this information was not well communicated in the current manuscript, we changed the text into:

“However, in combination with Ad, the initial downregulation of E2F1 recovered as early as 12 hours post-infection and was fully restored 36 hours later, while RB protein was even further suppressed (Fig. 4c). This indicates that Ad-related molecular factors regulate RB and E2F1 protein level even in the presence of CDK4/6i. It also indicates that this newly expressed E2F1 is activated since RB is not present to inhibit E2F at this situation.”

4. Comment 5 in the original review: The experiments using siE2F indicated that the E2F family were not decisive factor for the combination therapy in Figs. 5A and 5B. On the other hand, the replication activity of ADWT/E2Fm was much decreased compared to ADWT in Fig. 6D, which may indicate the importance of E2F1 in the virus replication. Please explain the reason.

We first want to comment on the siRNA effect in Fig 5A and 5B. We performed to these results a novel experiment in which we followed the time kinetic of E2F1, RB and E1A expression level.

Down-regulation of E2Fs by siRNA causes a reduction of around 70%. Interestingly, adenovirus infection strongly upregulates E2F gene expression and by that circumvent the effect of siRNA against E2F 24 hours past infection. We clarified this issue in the revised manuscript and included an additional figure (now figure 5d, former figure 5d-f are now e-g):

“In addition, adenovirus infection restores 24 hours past infection the siRNA induced suppression of E2F1 but suppresses RB, an effect that correlates with an enhanced expression level of E1A protein 12 hours past infection in E2F1 silenced cells (Fig 5d). This indicates that E2F1 has an important role at later time points in adenovirus replication.”

Based on these data, we do think that there might be a dual role for E2F1 in the replication cycle of an adenovirus. Our data indicate that there might be an early initiation of viral replication in which E2F might not be important and maybe even suppressive. As soon as E1A has been expressed, this situation seems to change and E2F1 level are despite the presence of CDK4/6 inhibitors or E2F1 directed siRNA stabilized, indicating that at this time point which is around 12 hours past infection the well established role of E2F1 as a transcription factor takes over. However, since at this

point the ADWT/E2Fm construct can not activate E2-expression, the decrease in replication can be explained.

Because this is a crucial aspect we decided to also clone a construct that has mutated E2F sites in both, the E1 enhancer and also the E2-early promoter (ADWT/2xE2Fm). We included the result as a novel Figure 7e and included into the text:

“Based on the results we thought that introducing a mutation of the E2F-binding sites at the E1A-enhancer (ADWT-2xE2Fm) would restore the inhibitory effect on E1A expression of the E2F-binding mutation in the E2-early promoter. As shown in Fig. 7e, while E2-early levels were still downregulated in ADWT-2xE2Fm cells, E1A levels were restored to ADWT level. Thus, the inhibitory effect of E1A expression caused by deletion of the two E2F-binding sites in the E2-early promoter is reversed by additional deletion of 1 E2F-binding at the E1A-enhancer region. These results support the hypothesis that different E2F binding sites within the viral genome compete for available E2F transcription initiation complexes in order to control and regulate viral replication. It also suggests, that the E2F binding sites within the E1-enhancer might act as negative regulatory elements.”

5. Comment 8 in the original review: Although the expression of E2 early gene may not be strongly correlated with the virus replication, it is still unclear why the E2 early promoter activity induced by ADWT and the replication of adenovirus were different between in pE2-early-luc-TrapM-transfected and pE2-early-luc-Trap-transfected cells (Figs. 6E and 6F). Since the readers would like to know the reason, please explain the reason to some extent.

We agree that the description of the replication was still not sufficiently explained. Thus, we changed the text into:

“In contrast, trapping of E2F/RB by transfecting cells first with the pE2-early-luc-Trap and infect then with adenovirus is beneficial for viral replication, supporting our findings with siRNAs and CDK4/6 inhibitors. In addition, the increase in adenoviral replication in cells that were transfected with the pE2-early-luc-TrapM treated with PD compared to pE2-early-luc-TrapM control also support the previous data because PD downregulates here the E2F/RB protein level (Fig.6f).”

6. In addition, the replication activity of ADWT/E2Fm and ADWT/Trap were opposite compared to ADWT in Figs. 6D and 6H. Is this reasonable?

Activation of the E2-early promoter depends mainly on the two E2F-binding sites within the promoter. A deletion of the binding sites present in ADWT/E2Fm will consequently reduce viral replication. Interestingly, we show that this vector results in accumulation of E2F at the E1A-Enhancer in the Chip assay in Fig.7c. This indicates that the different E2F binding sites compete for available E2F molecules in the cell.

In contrast, the addition of E2F bindings sites (ADWT/Trap) causes an increase in viral replication at least at low MOI, indicating that trapping of E2F/RB has a positive effect on replication. This result confirms now the results shown in figure 6e/f in a direct approach by cloning these sequences in a viral vector instead of using a plasmid.

7. Comment 9 in the original review: The authors did not show time course of viral replication in tumors and that CDK4/6 inhibitor enhanced the viral replication in the tumor xenograft of mice and the data of higher percentage of tumor cell degeneration and necrosis after combination therapy. It would be still valuable to show the results to confirm the enhanced virus replication in the tumor by the combination therapy.

We have not analyzed a time kinetic in xenografts. However, we now included additional in vivo results, confirming the results of enhanced replication, viral gene expression, protein expression of Hexon and E1A and downregulation of RB protein. We are aware that also these novel data do not fully address the question raised by the reviewer. However, we confirm higher replication level in xenografts in the combination vs the monotherapy reflecting our results from cell culture and the in vivo model in Fig 8. We updated Figure 8 and included a novel supplement figure 6.

The text in the manuscript was changed accordingly for Figure 8 and the novel supplementary figure 6.

8. Comment 11 in the revised version: In this study, the authors showed that the E2F1 is not a crucial factor and RB is important for the mechanism of the combination therapy. Please show the more detailed mechanism clarified in this study including the role of CDK4/6 inhibitors on RB and E2F1 in Figure 9.

We would like to point out that also in this manuscript as in many others, the role of E2F has not been finally clarified. We think that our data indicate that excess level of E2F are not beneficial for initial transcription of adenoviral genes but also indirectly show that E2F1 upregulation indicates that at some point there is a role of E2F in the viral life cycle. However, RB is a dominant regulator of E2F activity which might explain the effects we observe in the manuscript.

We have changed figure 9 and also the figure legend into:

“Adenovirus E1A and E2 expression is controlled by the well-balanced E2F/RB protein complex through E2F binding sites in their promoter region (black bars). Both promoter regions compete for binding of the E2F/RB protein complex indicating a novel level of regulatory mechanism in viral life cycle. Excess level of this protein complex suppress efficient transcription at early time points in the replication life cycle. Inhibition of RB and E2F1 protein level by CDK4/6-inhibitors might cause a shift towards low level of active E2F1 and thus facilitate viral gene expression and replication.”

Reviewer #3 (Remarks to the Author):

The authors seem not to understand the importance and relevance of orthotopic models of sarcoma to their study. The study would have been much stronger and more relevant with an orthotopic patient sarcoma mouse model (Oncotarget 7, 47556-47564, 2016; Cancer Letters 469, 332-339,2020). The authors' response to this point is unsatisfactory.

First, we completely agree with the reviewer that an orthotopic/patient-derived sarcoma

model is of great diagnostic and therapeutic value. We obviously did not respond accordingly to the raised question and would like to explain not only our issue with the mouse model asked for but also add data from a different mouse model to the manuscript.

First, to analyze the therapeutic effect of our virus-based approach it would in addition be necessary to establish a humanized patient derived sarcoma model, since the therapeutic effect of oncolytic viruses strongly depends on activating an anti-cancer immune response in vivo. We do not have such a sophisticated model established at this point and have applied for a grant addressing this question which of course is of very great importance to us.

In this context, we performed experiments with a second Ewing sarcoma cell line in a Rag2^{-/-} γc^{-/-} xenograft mouse model, in which tumor engraftment and growth is faster compared to the nude mouse model that has been performed and shown in the manuscript. In this novel model, we also observe a strong increase in viral genome replication, adenoviral E1A and hexon protein level with the combination therapy accompanied by reduced RB levels. In addition, a clear increase in caspase activity in the combination treatment is observed. We think that we can provide thus a stronger basis for our in vivo observation in our manuscript, which is the increase of adenoviral replication associated with RB down-regulation. Therefore, figure 8 was updated and we included a novel supplement 6.

We are aware of the fact that also these data cannot replace a humanized orthotopic mouse model but hope that these additional in vivo data are sufficient to show that our novel combination therapy is more effective than using only the oncolytic adenovirus as monotherapy. Also, we provide here now additional mechanistic insights that support our in vitro data.

REVIEWERS' COMMENTS

Reviewer #2 (Remarks to the Author):

This reviewer thinks that the authors appropriately responded to the comments.

Reviewer #3 (Remarks to the Author):

The subcutaneous tumor model of Ewing's sarcoma is not clinically relevant.

Response to Reviewer

REVIEWERS' COMMENTS

Reviewer #2 (Remarks to the Author):

This reviewer thinks that the authors appropriately responded to the comments.

Reviewer #3 (Remarks to the Author):

The subcutaneous tumor model of Ewing's sarcoma is not clinically relevant.

As we pointed out before, we agree with this statement and explained in detail why we have not included a clinically better suited mouse model at this point. However, the purpose of the whole study is to find novel ways to improve the oncolytic potency of XVir-N-31.

Monotherapy with oncolytic viruses is never very effective in 3-dimensional murine models. To show significant changes, one has to inject virus several times and in many publications 5 injections and more are applied.

In our novel combination therapy, we show that only two injections of virus are already sufficient to dramatically increase the oncolytic potency of XVir-N-31 in a 3-dimensional model and this is a clinically relevant observation.

Since it has been demonstrated extensively that it is the virus induced immune response that is crucial for a successful systemic therapy response, an efficient lysis of the tumor lesion will probably also induce a better recognition by the immune system. Since the lytic capability of an oncolytic virus in solid tumors is limited, this might explain the often-disappointing results in early clinical trials. Thus, our mouse model has a clinical relevance in that it provides evidence that the novel combination therapy increases the oncolytic potency in solid tumors.

We included in the discussion the following text:

The xenograft models used in this manuscript proof that also in a 3-dimensional in vivo situation the combination therapy used results in a much better oncolytic potency which corresponds to better replication compared to the virus monotherapy. However, for clinical purposes a humanized patient derived xenograft model would be beneficial since the therapeutic effect of oncolytic viruses strongly depends on activating an anti-cancer immune response in vivo (Mantwill et al., 2021).